# Every-other-day feeding extends lifespan but fails to delay many symptoms of aging in mice

Kan Xie et al.[#]

Dietary restriction regimes extend lifespan in various animal models. Here we show that longevity in male C57BL/6J mice subjected to every-other-day feeding is associated with a delayed onset of neoplastic disease that naturally limits lifespan in these animals. We compare more than 200 phenotypes in over 20 tissues in aged animals fed with a lifelong every-other-day feeding or ad libitum access to food diet to determine whether molecular, cellular, physiological and histopathological aging features develop more slowly in every-other-day feeding mice than in controls. We also analyze the effects of every-other-day feeding on young mice on shorter-term every-other-day feeding or ad libitum to account for possible aging-independent restriction effects. Our large-scale analysis reveals overall only limited evidence for a retardation of the aging rate in every-other-day feeding mice. The data indicate that every-other-day feeding-induced longevity is sufficiently explained by delays in life-limiting neoplastic disorders and is not associated with a more general slowing of the aging process in mice.

D.E. (email: Dan.Ehninger@dzne.de). [#]A full list of authors and their affliations appears at the end of the paper.

Aging is a major risk factor for most adult-onset disorders and is associated with a broad range of functional impairments[1]. Targeting aging processes with suitable pharmacological or dietary approaches could potentially represent a powerful inroad for the development of preventatives for aging-associated disorders[2].

The majority of studies performed to date to identify novel anti-aging interventions employed lifespan measures as a primary readout to detect effects on aging. Dietary restriction (DR) regimens are among the most extensively documented experimental interventions with longevity effects in a number of different species, including various invertebrate (e.g., *Caenorhabditis elegans* and *Drosophila melanogaster*), as well as vertebrate species (such as *Rattus rattus*, *Mus musculus*, and *Bos taurus*)[3].

Longevity effects have been demonstrated for different DR paradigms, including caloric restriction (CR)[4–7] and restriction regimens associated with intermittent periods of fasting, such as every-other-day feeding (EOD)[8, 9]. In the context of CR experiments, restricted animals are provided with a specified proportion (e.g., 60%) of the amount of food consumed by animals with ad libitum access to food (AL, i.e., by the control group). In intermittent fasting (IF) paradigms, food is provided generally AL to both controls and restricted mice, but restricted mice are, unlike controls, subjected to intermittent periods of fasting (IF); one specific IF protocol is EOD that alternates 24-h periods of full starvation and AL access to food. EOD extends lifespan without a substantial decrease of average daily food intake[10]. While life-extending effects of CR are correlated positively with the degree of food deprivation (up to 60%)[7], EOD extends average lifespan between 12 and 27% in mice if initiated early in life[7, 9]. The effects of CR on lifespan and age-related changes have been extensively studied over the past decades[3, 11], but much less attention has been paid to EOD.

In order to systematically study EOD's effects on aging and lifespan, we performed a large-scale analysis of >200 molecular, cellular, histopathological, and physiological parameters across >20 different tissues in mice subjected to EOD or AL access to food for much of their lifetime. The parameters selected corresponded in large part to the ones used in our prior large-scale analysis of rapamycin's effects on aging in male C57BL/6J mice[12]. In addition to the analyses of aging phenotypes, we determined EOD's effects on lifespan and assessed causes of natural death in cohorts chronically subjected to EOD or AL.

Our data show that lifespan extension induced by EOD in male C57BL/6J mice is sufficiently explained by a delay in life-limiting neoplastic disorders. Our large-scale analysis of molecular, cellular, physiological, and histopathological aging phenotypes identified only few aging features delayed by EOD, indicating that the observed extension in lifespan was not associated with a more general slowing of aging processes in EOD mice.

## Results

### EOD extended lifespan and delayed lethal neoplastic disease.
Over the course of their entire lifespan, average calorie intake of EOD mice was reduced by 7.5% (controls: 11.06 kcal/day; EOD: 10.23 kcal/day) (Fig. 1b). Body weights remained reduced in EOD mice during much of their adult life (average reduction of ~17.1%) (Fig. 1c). Organ weights and body dimensions were often reduced in EOD mice compared to AL controls (Supplementary Fig. 1). Consistent with prior findings[9], we observed significant lifespan extension in EOD mice (Fig. 1d). Mean lifespan was prolonged by 102 days (AL: 806 days, EOD: 908 days) and maximum lifespan, calculated based on the longest living 20%, was extended by 122 days (AL: 980 days, EOD: 1102 days) (Fig. 1e), which is within the range of lifespan extension observed

in previous DR studies[13]. Calculation of mortality doubling time using the Gompertz function revealed no substantial difference between AL and EOD animals (AL: 83.6 days, EOD: 91.7 days). A detailed pathological examination of dead mice revealed that most EOD and AL mice had died of neoplastic disorders (Fig. 1f), most notably hematological malignancies, known leading natural causes of death in both sexes of many inbred mouse strains, including C57BL/6J[6, 14]. Though cancer remained unchanged as the primary cause of death, tumor burden at death was significantly reduced in EOD mice compared to controls (average number of tumors per mouse: AL = 1.28, EOD = 0.85) (Fig. 1g and Supplementary Table 1). The findings indicate that EOD mice lived longer than controls because lethal neoplastic disease was delayed in these animals.

### Large-scale phenotyping to assess EOD effects on aging.
In order to systematically study EOD's effects on aging and lifespan, we performed a large-scale analysis of >200 molecular, cellular, histopathological, and physiological parameters across >20 different tissues in mice subjected to EOD or AL access to food for much of their lifetime. To be able to account for acute EOD effects, our study design included young groups of mice subjected to EOD or AL for only 1 month prior to the commencement of phenotypic analyses at 3 months of age (Fig. 1a).

On the basis of their modulation by age (old vs. young) and/or diet (EOD vs. AL), traits from the large-scale phenotypic analyses were categorized as follows: (i) aging phenotypes prevented or delayed by EOD (Supplementary Table 2). These were aging phenotypes that were specifically rescued by EOD in the aged group of mice kept on almost lifelong EOD but that remained unaltered in young mice on short-term treatment; (ii) age-sensitive parameters opposed by EOD with similar outcomes in young and old mice (Supplementary Table 3). Similar EOD effects on a trait in both young mice on short-term treatment, and aged mice subjected to EOD for much of their lifetime, cannot be taken as evidence that EOD prevented or delayed aging; (iii) aging phenotypes not measurably altered or worsened by EOD (Supplementary Table 4); (iv) parameters modulated by EOD but not by age (Supplementary Table 5); (v) traits neither modified by aging nor by EOD. Experimental findings regarding aging-associated phenotypes are described in the remainder of the "Results" section, in Figs. 2–7, Supplementary Tables 6–17, as well as Supplementary Data sets 1 and 2. A summary of the findings is provided in the concluding section of the "Results" section and in Fig. 1h–j.

### EOD had age-independent effects on neurological phenotypes.
We started our analysis of age-related health parameters by testing neurological and behavioral functions in aged mice subjected to almost lifelong EOD or AL access to food, as well as young mice on short-term EOD or AL. If not stated otherwise, all measurements were initiated in the morning after a feeding day. A modified SHIRPA screen was used to assess general health, autonomic functions, reflexes, posture, movement, and spontaneous behavior. Consistent with prior observations[12], we found age-related changes on a subset of SHIRPA measures (presence of tremor, gait abnormalities, tail elevation, reduced vocalization, reduced, or absence of startle response) (Supplementary Table 6). These did not appear to be prevented by EOD. We do note, however, that larger sample sizes would be needed to be able to detect effects on these types of measures.

Exploratory activity was examined in an open-field assay, in which we allowed mice to explore freely a novel environment for a 20-min period. Consistent with previous observations[12, 15], these experiments showed overall reduced exploration in aged mice compared to young animals, as indicated by reduced

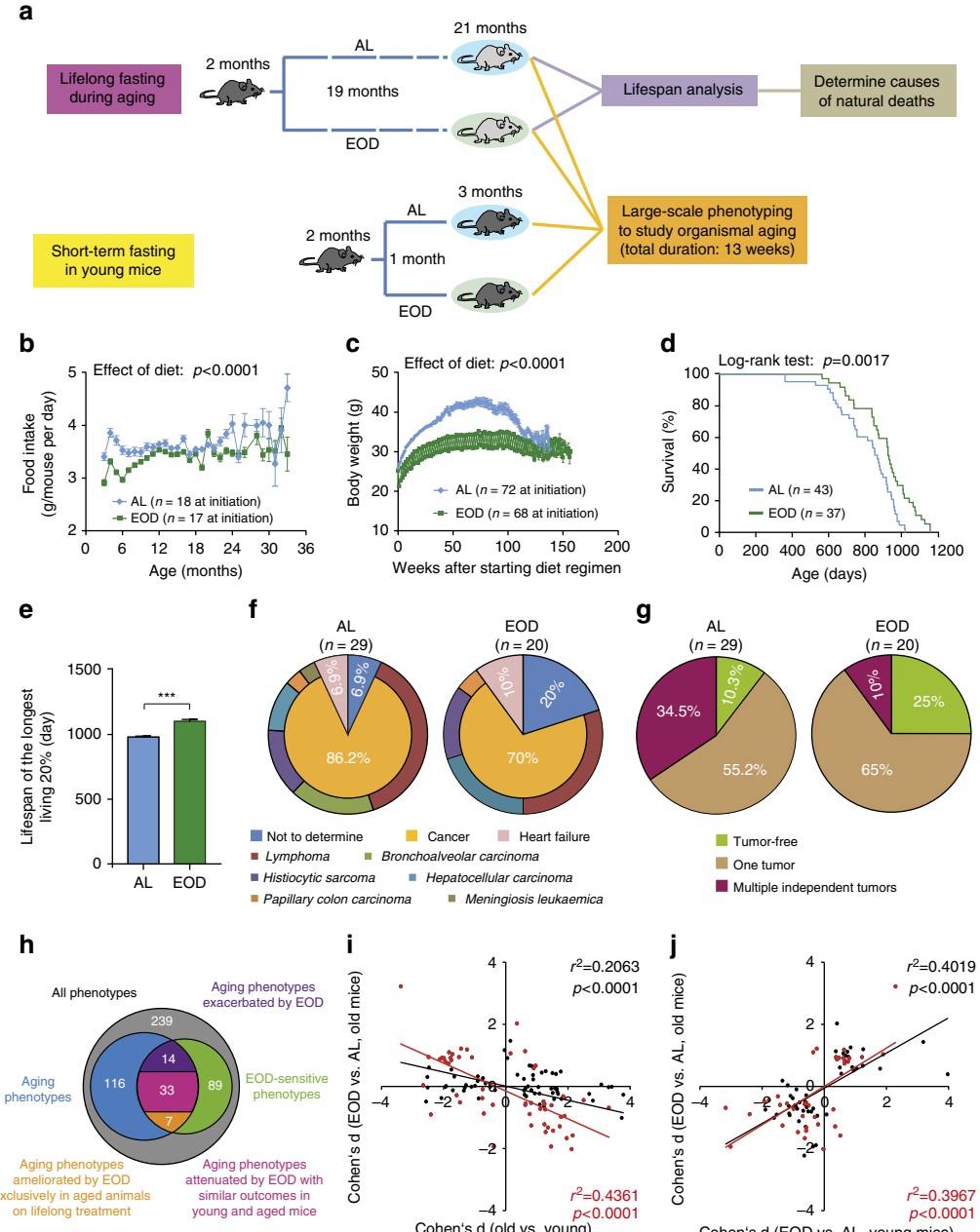

**Fig. 1** EOD extended lifespan and delayed lethal neoplastic disease but had limited effects on aging rate. **a** Schematic illustration of the experimental design. **b** Estimated food intake per animal (number of cages at initiation AL: $n = 18$, EOD: $n = 17$). **c** Body weight (AL: $n = 72$ mice, EOD: $n = 68$ mice at initiation). Statistical analyses for **b** and **c** were performed using a mixed effects model with DR as fixed effect and cage and time as random effects. **d** Survival curves (log-rank test; AL: $n = 43$ mice, EOD: $n = 37$ mice). **e** Maximum lifespan was calculated from the longest living 20% (t-test; AL: $n = 9$ mice, EOD: $n = 8$ mice). **f** Causes of natural deaths in AL ($n = 29$ mice) and EOD ($n = 20$ mice) animals. **g** Tumor burden in animals that died of natural causes (AL: $n = 29$ mice, EOD: $n = 20$ mice). **h–j** Summary of large-scale phenotyping analyses to study organismal aging in AL and EOD mice. **h** The Venn diagram provides an overview regarding the total number of phenotypes assessed ($n = 239$), the number of aging traits ($n = 116$), and parameters modified by EOD ($n = 89$), as well as overlaps of these sets of measures. Forty aging traits were ameliorated by EOD, of which 7 were prevented by EOD and 33 were modulated with similar outcomes in young mice, treated for only 1 month, and in aged animals kept on almost lifelong EOD. Fourteen aging traits were exacerbated by EOD. **i** Effect sizes (Cohen's d) of age are plotted against effect sizes of EOD (in old mice) for all aging traits (*black dots*; $n = 116$), as well as for all traits modulated by both age and dietary intervention (*red dots*; $n = 53$). In both cases, the data show an inverse correlation of the effects of age and DR. **j** Effect sizes (Cohen's d) of EOD in young mice are plotted against effect sizes of EOD in old mice for all traits modulated by diet (*black dots*; $n = 88$), as well as for all aging traits opposed by EOD (*red dots*; $n = 39$). In both cases, the data show a correlation of EOD effect sizes in young and old mice. *AL* ad libitum, *EOD* every-other-day feeding

distance traveled measures and reduced numbers of rearings (Fig. 2a, b). EOD increased exploratory activity in both young mice on short-term treatment and aged mice on almost lifelong treatment (Fig. 2a–d; full data set in Supplementary Table 7).

We examined motor coordination and balance using the accelerating rotarod. These analyses showed the expected[12, 15] age effects with significantly reduced latencies to fall in aged animals (Fig. 2f). We observed a possible slight improvement of latencies

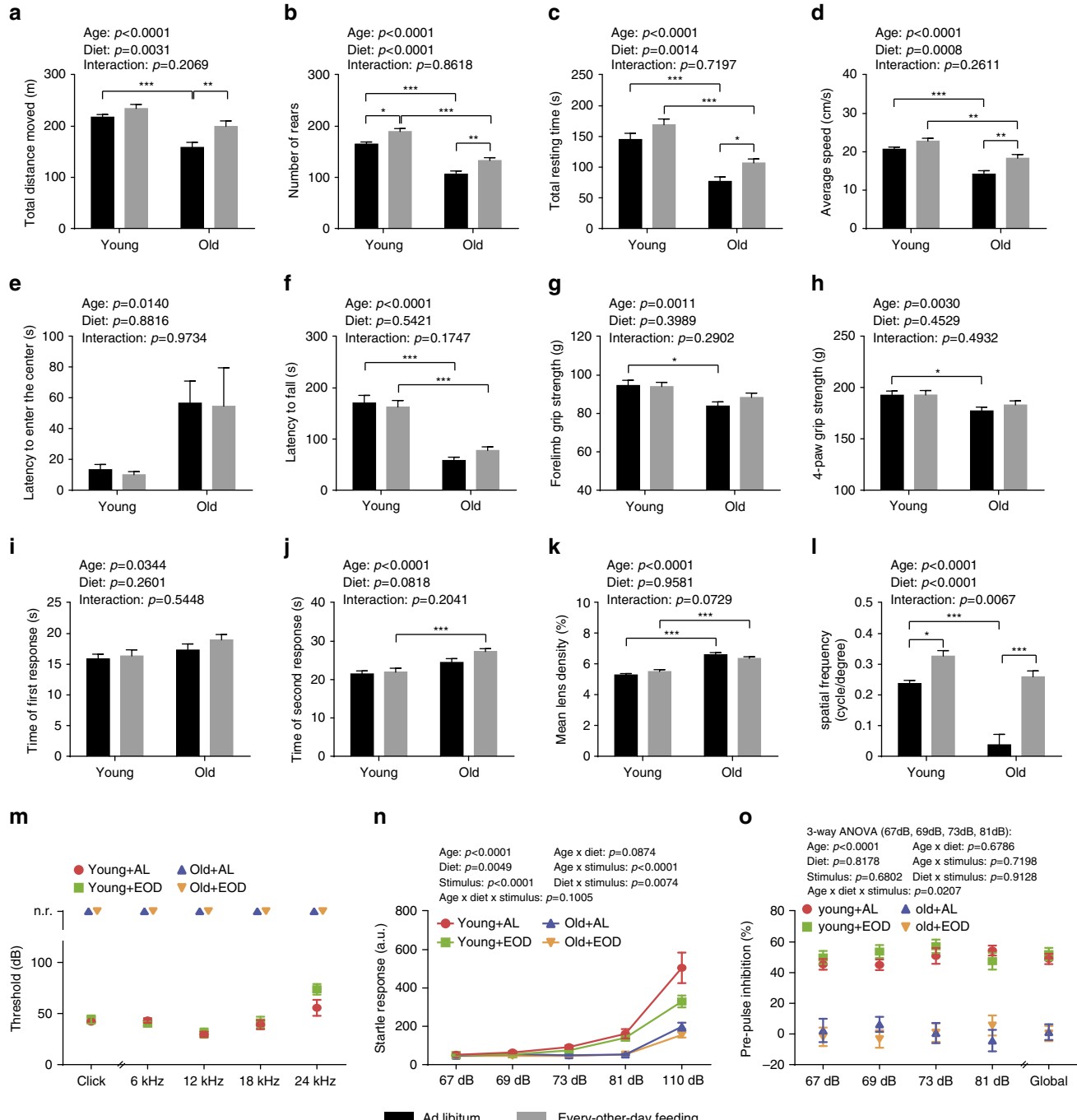

**Fig. 2** Neurological and ophthalmological phenotypes. **a–j** Assessment of neurological and behavioral phenotypes (two-way ANOVA; young + AL: $n = 16$, young + EOD: $n = 16$, old + AL: $n = 23$, old + EOD: $n = 23$). Analysis of locomotor activity including **a** total distance moved, **b** number of rears, **c** total resting time, **d** average speed, and **e** latency to enter the center as determined in an open-field test. **f** Latencies to fall on an accelerating rotatod. **g**, **h** Muscle strength measured by two-paw (forelimb) and four-paw grip strength tests. **i**, **j** Latencies to first and second reaction in a hot-plate-based assessment of nociceptive function. **k** Quantification of mean lens density by Scheimpflug imaging (two-way ANOVA; young + AL: $n = 7$, young + EOD: $n = 8$, old + AL: $n = 7$, old + EOD: $n = 6$). **l** Assessment of visual acuity using the virtual drum test (two-way ANOVA; young + AL: $n = 7$, young + EOD: $n = 8$, old + AL: $n = 7$, old + EOD: $n = 6$). Analysis of **m** auditory brain stem responses (young + AL: $n = 12$, young + EOD: $n = 12$, old + AL: $n = 10$, old + EOD: $n = 10$), **n** acoustic startle response and **o** pre-pulse inhibition (PPI). Statistical analyses for **n** and **o** were performed using a three-way ANOVA with the between-subjects factors age (young vs. old), DR (AL vs. EOD), and the within-subjects factor sound intensity (Greenhouse-Geisser correction) (young + AL: $n = 16$, young + EOD: $n = 16$, old + AL: $n = 23$, old + EOD: $n = 23$). *AL* ad libitum, *EOD* every-other-day feeding, *a.u.* arbitrary unit, *n.r.* no response. Data are shown as mean ± SEM. Full data sets of the open-field test and Scheimpflug imaging are presented in Supplementary Tables 7 and 8

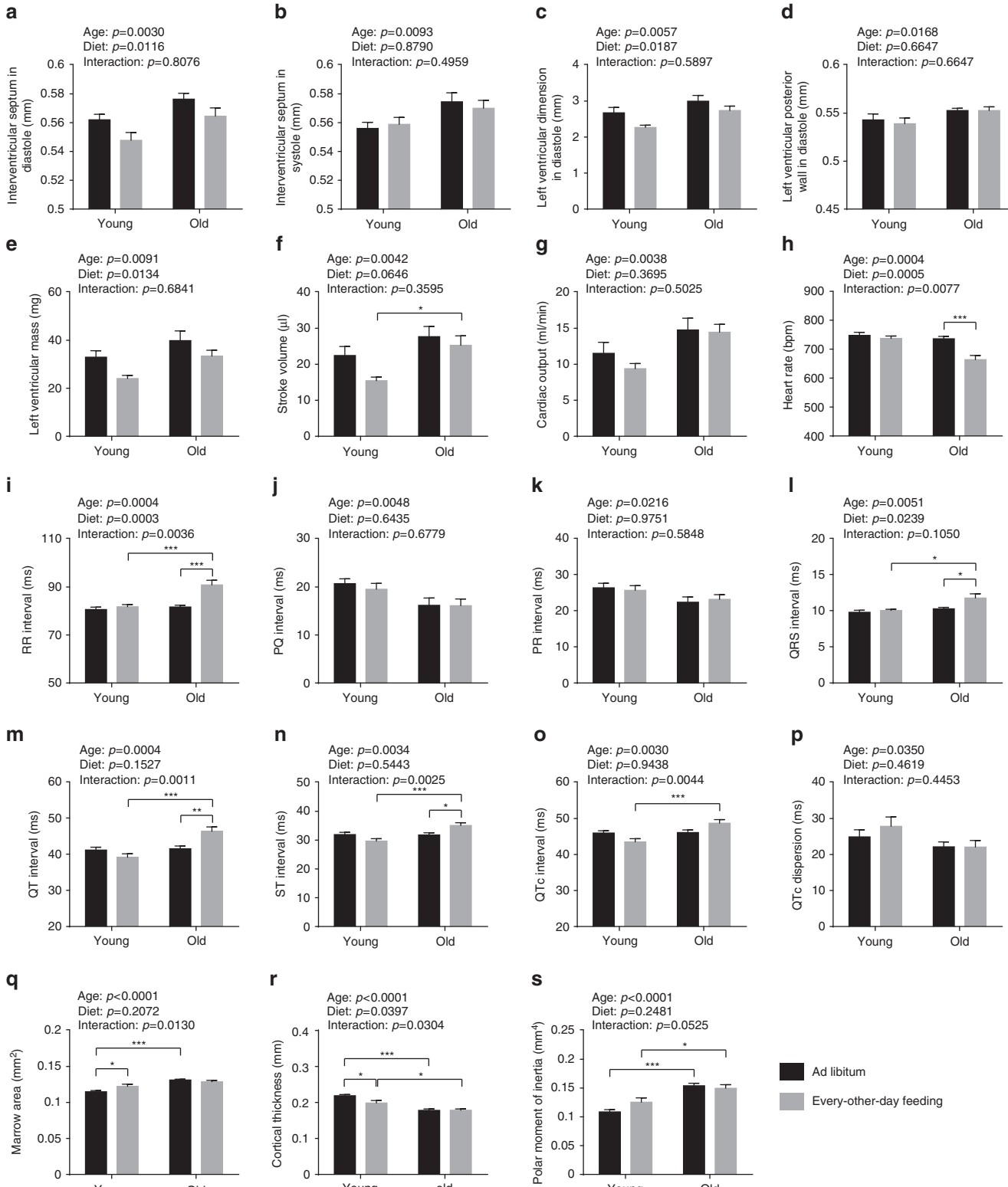

**Fig. 3** Cardiological analyses and assessment of bone structure. **a–g** Ventricular dimensions and functions were examined with transthoracic echocardiography (two-way ANOVA; young + AL: $n = 10$, young + EOD: $n = 9$, old + AL: $n = 11$, old + EOD: $n = 11$). Dimensions of **a** interventricular septum in diastole, **b** interventricular septum in systole, **c** left ventricle in diastole, and **d** left ventricular posterior wall in diastole were determined. **e** Mass of left ventricle, **f** stroke volume, and **g** cardiac output were calculated. **h–p** ECGs were recorded from conscious mice (two-way ANOVA; young + AL: $n = 10$, young + EOD: $n = 9$, old + AL: $n = 11$, old + EOD: $n = 11$). **h** HR, **i** RR interval, **j** PQ interval, **k** PR interval, **l** QRS interval, **m** QT interval, **n** ST interval, **o** QTc interval, **p** QTc dispersion. **q–s** Bone architecture was examined using micro-CT of distal tibia (two-way ANOVA; young + AL: $n = 16$, young + EOD: $n = 9$, old + AL: $n = 17$, old + EOD: $n = 9$). **q** Marrow area, **r** cortical thickness, **s** polar moment of inertia. *AL* ad libitum, *EOD* every-other-day feeding. Data are shown as mean ± SEM. Full data sets of echocardiography, electrocardiography, and micro-CT assessments are presented in Supplementary Tables 10–12

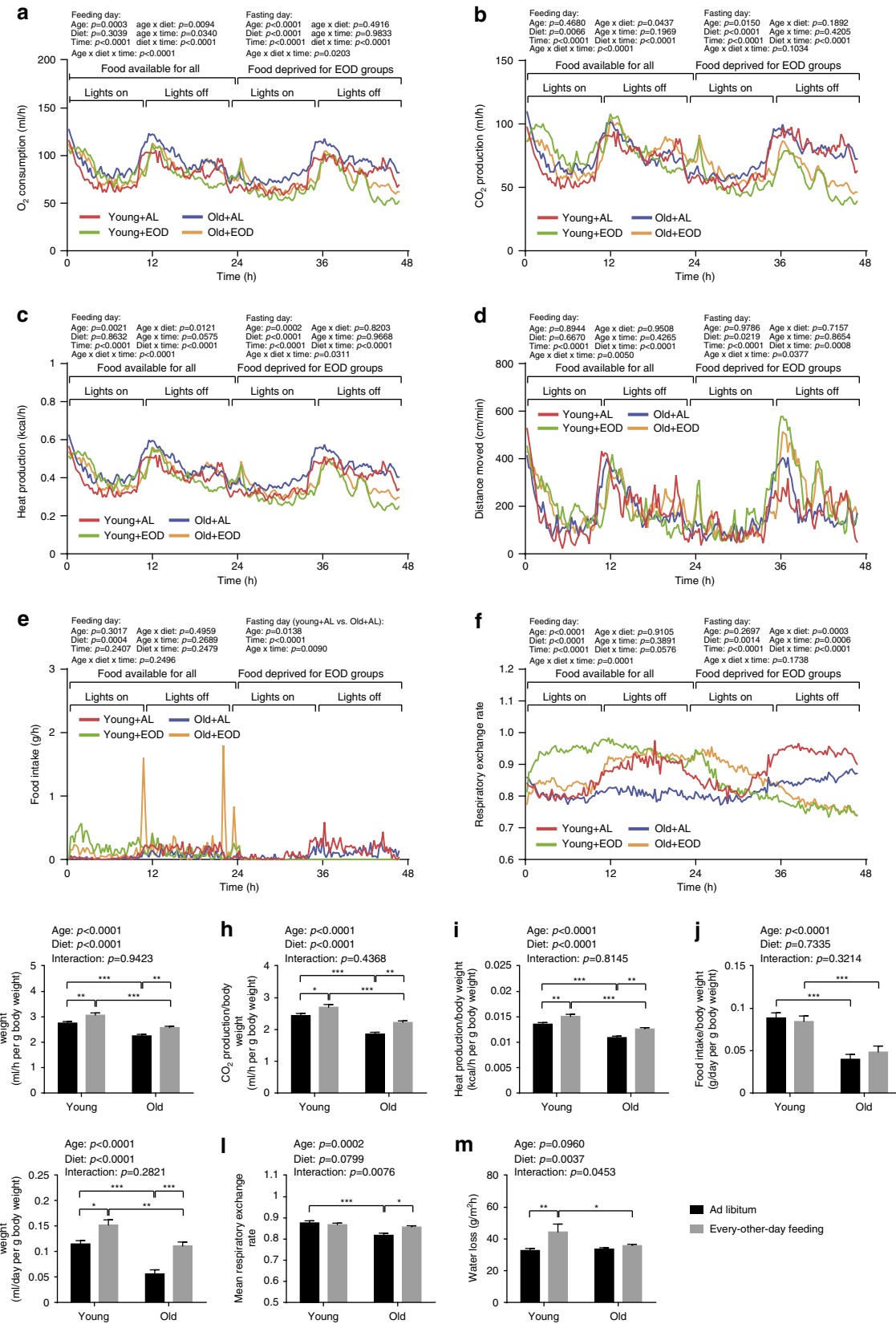

to fall in EOD-treated old mice compared to age-matched controls but without reaching statistical significance (Tukey post hoc test following two-way ANOVA, old/AL vs. old/EOD: $p = 0.4146$) (Fig. 2f). We also examined motor strength in EOD

mice and controls using two-paw and four-paw versions of a grip strength test. These analyses revealed the expected[12, 15] age-related reductions in grip strength and suggested a small but statistically insignificant improvement in EOD-treated old mice

relative to AL-fed aged controls (Tukey post hoc tests following two-way ANOVAs; forelimb grip strength, old/AL vs. old/EOD: $p = 0.4463$, four-paw grip strength, old/AL vs. old/EOD: $p = 0.6789$) (Fig. 2g, h).

We assessed nociceptive functions in EOD mice and controls using a hot plate test. These studies showed the expected[12] increases in reaction latencies in aged mice compared to young animals, suggesting age-related alterations in nociceptive function. EOD did not attenuate these aging-associated behavioral changes in the hot plate test but instead further increased reaction latencies in aged mice (Fig. 2i, j).

**EOD did not alter age-related increases in lens density.** The most common aging-associated pathology in the anterior part of the eye is cataract formation[16]. We used Scheimpflug imaging to measure abnormalities in the anterior segment of the eye, including changes in lens density in young and old EOD and control mice. These studies showed the expected[12, 16] age-related increase in mean lens density (Fig. 2k). EOD did not improve aging-associated increases in mean lens density in any obvious and measurable way (Fig. 2k; full data set in Supplementary Table 8).

The posterior part of the eye was assessed using in vivo optical coherence tomography, fundoscopy, and histological analyses, demonstrating, consistent with previous observations[12], normal retinal thickness and apparently unaltered retinal vasculature in aged mice (Supplementary Table 9).

Visual acuity was examined using the virtual drum test. Consistent with results obtained from previous work[12, 17], performance of aged animals was worse compared to young animals (Fig. 2l). Only one out of seven aged control mice was able to complete this task, whereas performance was improved in old EOD animals. EOD enhanced performance in both young and aged cohorts, indicating age-independent effects on performance in this visual acuity task (Fig. 2l).

**Age-dependent hearing loss was not ameliorated by EOD.** Presbycusis is a common issue in elderly individuals and is associated with a loss of hair cells and spiral ganglion sensory neurons[18]. Age-dependent hearing loss also is a feature of C57BL/6J mice, beginning between 12 and 15 months of age in the higher frequencies and progressing to lower frequencies[18]. For a gross assessment of age-dependent hearing loss in young and old control and EOD mice, we subjected animals to the clickbox test, in which the responsiveness of mice to a 20 kHz sound is recorded. These experiments showed the expected[12, 18] lack of responding in the aged groups (Fig. 2m). Auditory brain stem response measurements confirmed the complete aging-associated hearing loss and showed no EOD effects on stimulus intensity thresholds (Fig. 2m). In summary, aged animals showed the expected sensorineural hearing loss, which was unaffected by

EOD at that age. However, intermediate shift of onset or progression was not assessed.

We also assessed acoustic startle reactivity and pre-pulse inhibition (PPI) in young and old EOD mice and controls. As expected[12, 19], aged animals showed reduced startle responses to a range of different sound intensities (Fig. 2n). EOD decreased responding independent of age and across several sound intensities (Fig. 2n). PPI was largely absent in the aged groups (Fig. 2o). EOD in young mice had no effect on PPI (Fig. 2o). Together, these data are consistent with the clickbox data and auditory brain stem response measurements mentioned above and indicate that age-related hearing loss was not prevented by EOD.

**EOD reduced heart weight and cardiac dimensional measures.** Aging is associated with significant changes in heart structure and function[20]. To test for gross structural changes in the aged heart, we measured heart dimensions via echocardiography and obtained heart weights upon pathological assessment of the animals. Some dimensional measures showed expected[12, 21, 22] age-related increases that were either partially or fully restored by EOD with similar effects in young and old mice (Fig. 3a–e; full data set in Supplementary Table 10): thickness of interventricular septal wall in diastole (IVSd), left ventricular end-diastolic internal diameter (LVIDd), corrected left ventricular mass (LV mass corr). Consistent with these echocardiography findings, we observed reduced heart weights in EOD mice of both age groups upon pathological assessment (Supplementary Fig. 1i). EOD also altered a number of additional structural and functional echocardiography measures, including fractional shortening (FS), ejection fraction, and stroke volume (Fig. 3f and Supplementary Tables 10 and 11). These EOD effects often appeared to be more pronounced in the young group than in the aged set of animals. In sum, EOD effects were consistent with a retardation of heart growth (particularly evident in the young group) and consequently reductions in heart dimensional measures described above.

**EOD did not modify age-related changes in bones and joints.** Progressive bone loss with age results in osteoporosis, reduced bone strength, and fractures[23]. We measured cortical bone architecture by micro-CT analyses of the distal tibia. Consistent with previous reports[24, 25], significant age-dependent alterations were detected in a subset of parameters (decreased cortical thickness with increased marrow cavity area and polar moment of inertia) (Fig. 3q–s; full data set in Supplementary Table 12). Age-dependent changes in cortical bone architecture were not affected by EOD. We also performed X-ray radiography of spine, ribs, scapulae, clavicle, pelvis, femur, humerus, ulna, tibia, digits, joints, jaws, teeth, fibula, radius, ribs, and skull, confirming no obvious skeletal abnormalities in aged or EOD mice detectable at this level of analysis.

**Fig. 4** Energy metabolism and water turnover. **a–l** Indirect calorimetry was conducted over a period of 47 h, initiated at 7 a.m. (young + AL: $n = 16$, young + EOD: $n = 16$, old + AL: $n = 16$, old + EOD: $n = 16$). The dark/light cycle was kept at 12:12 h (lights on 6 a.m. CET, lights off 6 p.m. CET). For mice assigned to the EOD cohorts, food was removed after the first 24 h. Water was supplied AL to all groups. Temporal pattern of **a** oxygen consumption, **b** carbon dioxide production, **c** HP, **d** distance moved, **e** food intake, and **f** respiratory exchange rate are shown. **g** Oxygen consumption, **h** carbon dioxide production, **i** HP, **j** food intake, and **k** water consumption per g body weight as well as **l** mean respiratory exchange rate were calculated. **m** TEWL was measured non-invasively using a special Tewameter (young + AL: $n = 15$, young + EOD: $n = 16$, old + AL: $n = 22$, old + EOD: $n = 23$). Statistical analyses of **a–f** were performed via three-way ANOVA with the between-subjects factors age (young vs. old), DR (AL vs. EOD) and the within-subjects factor time (Greenhouse-Geisser correction) for feeding and fasting day, respectively. Statistics of **g–m** were calculated via two-way ANOVA with between-subjects factors age (young vs. old) and DR (AL vs. EOD). AL ad libitum, EOD every-other-day feeding. Data in **a–f** are presented as mean. Data in **g–m** are shown as mean ± SEM. The full indirect calorimetry data set is presented in Supplementary Table 13

**Calorimetry revealed altered metabolic profiles in EOD mice.** Indirect calorimetry was conducted over a period of 47 h, such that, in the EOD groups of mice, time windows with AL access to food (24 h), as well as complete food restriction (23 h) were covered experimentally. During the day with AL access to food, all groups (i.e., young/old, AL/restricted) showed a typical ultradian rhythm in energy expenditure and physical activity with lower gas exchange during the day and an increase in the evening triggered by the onset of the dark phase (Fig. 4a–d). On the food restriction day, daytime energy expenditure did not differ considerably between groups (Fig. 4c). EOD mice, however, tended to show reduced oxygen consumption and carbon dioxide

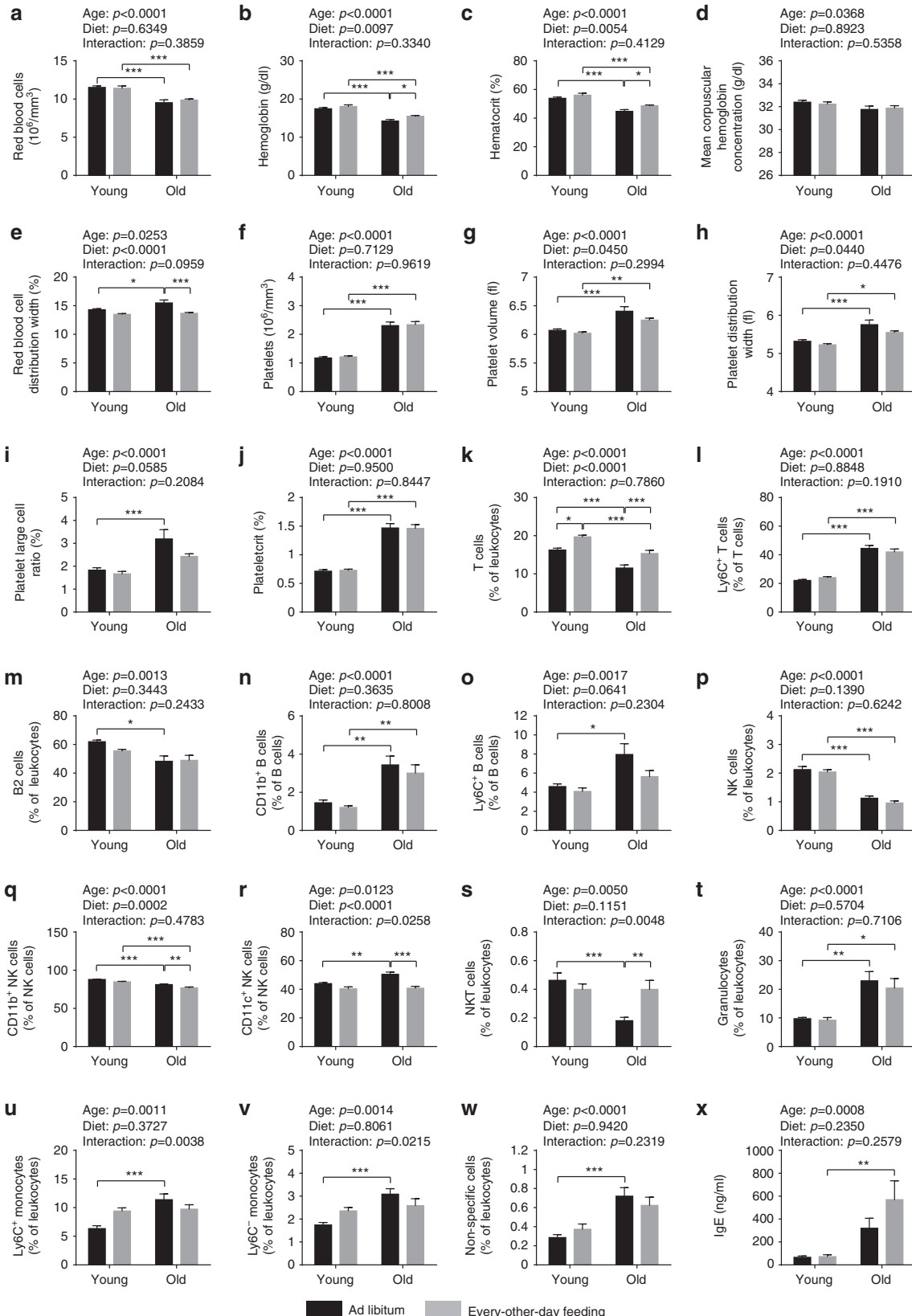

production rates during the second half of the night of the food restriction day (Fig. 4a, b). Only two restricted mice (one young, one aged) showed a metabolic suppression similar to torpor typical for small rodents. Overall, levels of oxygen consumption, carbon dioxide production, and heat release, however, significantly increased in EOD mice when body weight differences were accounted for (Fig. 4g–i; full data set in Supplementary Table 13).

The availability of food as well as the age of the mice had effects on metabolic fuel utilization (Fig. 4e, f, l). Consistent with prior findings[12, 26], aged control mice showed significantly reduced average respiratory exchange ratios (RER) compared to young control animals (Fig. 4f, l), indicating a proportional substrate preference toward lipid oxidation in aged mice. The metabolic profile of young control mice was dominated by low RER (indicative for lipid oxidation) during daytime and shifted to higher RER (reflective for carbohydrate utilization) in the evening (Fig. 4f). Aged control mice, in contrast, showed low RER values with little variability throughout the entire period of assessment (Fig. 4f). Young EOD mice showed high RER values during the feeding day and low RER values during the starvation day (Fig. 4f). The temporal RER profile in old EOD mice showed similarities with that in young EOD mice but was shifted by ~12 h, such that RER began to rise at ~12 h after onset of food availability and remained high for ~12 h during the starvation day (Fig. 4f). The RER profiles in the young groups described above were consistent with the observation that food intake occurred primarily during the dark phase in controls, while young EOD mice also engaged in food consumption during the light phase (Fig. 4e). This pattern appeared to be less pronounced in aged animals on almost lifelong EOD (Fig. 4e), possibly reflecting adaptation to this long-term feeding regimen.

**Increased water turnover in EOD mice.** Transepidermal water loss (TEWL) measurements are useful as an initial approach for identifying skin damage (e.g., associated with pathological conditions, such as eczema, or caused by chemical or physical insults), whereby TEWL rates increase proportionally to the level of damage. We subjected aged and young EOD and control mice to TEWL measurements using a Tewameter placed on the skin. Our analyses showed no obvious effect of aging but an overall increase of TEWL in EOD mice relative to AL-fed controls (Fig. 4m). Consistent with increased rates of water loss in EOD mice, measurements of water intake showed excessive water consumption in these animals (Fig. 4k and Supplementary Table 13). It remains to be determined in future studies whether excessive water intake serves to compensate for water loss or whether there is increased TEWL due to increased rates of water uptake. One likely scenario may be that excessive food intake during the feeding day is associated with increased water intake and consequently increased water loss in EOD mice.

**Altered body composition in EOD mice.** We examined body composition (total fat, lean mass) in young and old EOD, as well as AL-fed animals using time domain nuclear magnetic resonance (TD-NMR). Advanced age was associated with the expected[12, 26] increase in proportional body fat mass (fat mass adjusted to body weight), as well as decrease in relative lean mass. Body mass was consistently decreased in EOD mice of both age cohorts. Of note, body composition was altered in EOD mice with a slight increase in proportional fat mass and a corresponding decrease in relative lean mass (Supplementary Fig. 1a–c), which likely reflects an adaptive response to the intermittent availability of food.

**EOD reduced leukocyte counts and increased hematocrit.** Aging in C57BL/6J mice is known to be associated with an altered cellular composition of the peripheral blood[12, 27, 28]. Therefore, we included in our analysis a basic hematological assessment of young and old EOD mice and controls. Our observations, outlined below, are consistent with known aging effects on blood cell counts (Fig. 5; full data set in Supplementary Table 14)[12, 27, 28]. Red blood cell counts (RBC) were decreased in aged mice (Fig. 5a). Consistent with the RBC results, hemoglobin concentrations and hematocrit (HCT) were reduced in aged mice (Fig. 5b, c). RBC distribution width (RDW) values were increased in aged animals (Fig. 5e), indicating a higher degree of anisocytosis. Although EOD left RBC unaffected, it caused an overall increase in hemoglobin concentrations and HCT across age groups (Fig. 5b, c), accompanied by an increase in mean corpuscular volume (MCV) and mean corpuscular hemoglobin content (Supplementary Table 14). In addition, EOD reduced RDW values in both the young and aged group of mice (Fig. 5e). Overall, similar treatment outcomes in young and aged animals suggested that these changes in hematological parameters were related to age-independent effects associated with EOD. Our studies also revealed elevated platelets counts in aged animals (Fig. 5f). Though no measurable EOD effect on platelet counts and plateletcrit (PCT) was detected, platelet volume, platelet distribution width (PDW), and platelet large cell ratio were reduced in EOD animals regardless of age (Fig. 5f–j). Leukocyte counts were reduced by EOD across age groups (Supplementary Table 14).

**EOD attenuated a subset of aging-associated immune changes.** Age-related immune dysfunction leads to an increased prevalence of infections and autoimmune disease in the elderly and may also contribute to the increased cancer risk in aged individuals[29, 30]. Immune aging phenotypes affect the lymphocyte, as well as myeloid compartment in the peripheral blood and include decreased numbers of T and B lymphocytes, decreased natural killer (NK) cell frequencies, as well as increased frequencies of monocytes and neutrophil granulocytes[29, 30].

To test for alterations with regard to these immune phenotypes, we performed ten-color polychromatic flow cytometry and quantified a subset of immune cell populations in young and old EOD and AL-fed mice. Consistent with published data[12, 31–34], these experiments showed aging-associated decreases in T cells, B2 cells, as well as NK cells (Fig. 5k, m, p;

**Fig. 5** Hematological and immunological analyses. **a–j** Number and size of the different blood cell types were determined using a hematology analyzer (two-way ANOVA; young + AL: $n = 16$, young + EOD: $n = 16$, old + AL: $n = 20$, old + EOD: $n = 22$). **a** RBC count, **b** hemoglobin concentration, **c** HCT, **d** MCHC, **e** RDW, **f** platelet count, **g** platelet volume, **h** PDW, **i** large cell ratio of platelets (>12 fl), and **j** PCT. **k–w** Peripheral blood leukocytes were analyzed using a ten-color flow cytometer (two-way ANOVA; young + AL: $n = 16$, young + EOD: $n = 16$, old + AL: $n = 20$, old + EOD: $n = 22$). Only CD45-positive cells were included in the analysis. **k** T cells (CD3$^+$CD5$^+$), **l** Ly6C$^+$ T cells, **m** B2 cells (CD5$^-$CD19$^+$B220$^+$), **n** CD11b$^+$ B cells (CD19$^+$B220$^+$), **o** Ly6C$^+$ B cells (CD19$^+$B220$^+$), **p** natural killer (NK) cells (CD3$^-$CD5$^-$NKp46$^+$ and/or NK1.1$^+$), **q** CD11b$^+$ NK cells, **r** CD11c$^+$ NK cells, **s** NKT cells (CD3$^+$CD5$^+$NKp46$^+$ and/or NK1.1$^+$), **t** granulocytes (CD11b$^+$Ly6G$^+$), **u** Ly6C$^+$ monocytes (CD11b$^+$), **v** Ly6C$^-$ monocytes (CD11b$^+$), **w** non-specific cells, **x** plasma concentration of IgE immunoglobulins. *AL* ad libitum, *EOD* every-other-day feeding. Data are shown as mean ± SEM. Full data sets of hematological and immunological analyses are presented in Supplementary Tables 14 and 15

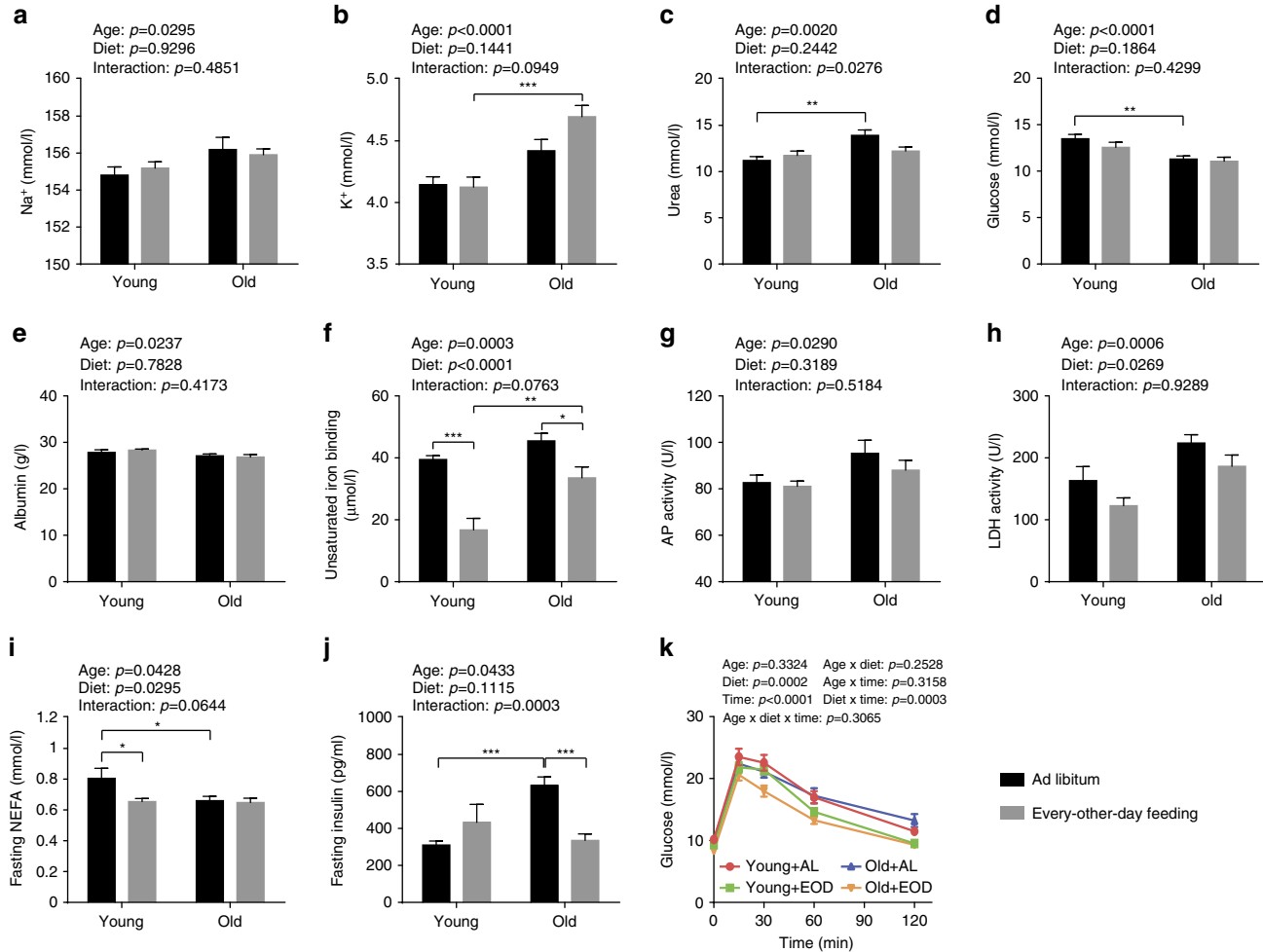

**Fig. 6** Clinical chemistry measures in plasma of fed and fasted mice. Plasma concentrations of **a** sodium, **b** potassium, **c** urea, **d** glucose, and **e** albumin, **f** unsaturated iron-binding capacity, as well as activities of **g** alkaline phosphatase (*AP*) and **h** lactate dehydrogenase (*LDH*) were determined in fed mice (two-way ANOVA; young + AL: $n = 16$, young + EOD: $n = 16$, old + AL: $n = 20$, old + EOD: $n = 22$). Plasma concentrations of **i** fasting NEFA and **j** fasting insulin were measured in animals after a 6 h fasting period (two-way ANOVA; young + AL: $n = 16$, young + EOD: $n = 16$, old + AL: $n = 23$, old + EOD: $n = 23$). **k** Blood glucose concentration during an IpGTT (three-way ANOVA with the between-subjects factors age (young vs. old), DR (AL vs. EOD), and the within-subjects factor time points (Greenhouse-Geisser correction); young + AL: $n = 16$, young + EOD: $n = 16$, old + AL: $n = 23$, old + EOD: $n = 23$). *AL* ad libitum, *EOD* every-other-day feeding. Data are presented as mean ± SEM. Full clinical chemistry data sets are presented in Supplementary Tables 16 and 17

full data set in Supplementary Table 15). EOD increased T-cell frequencies in both the young mice on short-term EOD and aged mice chronically subjected to EOD during aging, indicating age- and exposure-time-independent effects of EOD on this measure (Fig. 5k). Aging is known to be associated with an increase in the proportion of T cells expressing memory markers, such as CD44 and/or Ly6C[12, 35]. The age-dependent increase of Ly6C+ T cells, observed in the present study, remained unaffected by EOD (Fig. 5l). Neither expected age-related decreases of B2 cells[33] nor increases of CD11b+ and Ly6C+ B-cell frequencies were prevented by EOD (Fig. 5m–o). Furthermore, EOD had no measureable effect on the expected[12, 32] aging-associated decline in NK cell frequencies (Fig. 5p). However, NK cell subpopulations were modified by EOD, such that levels of CD11b+ and CD11c+ NK cells were reduced (Fig. 5q, r) and the percentage of Ly6C+ NK cells was elevated (Supplementary Table 15). Interestingly, age-dependent decreases of NKT cells were fully recovered by EOD (Fig. 5s). With regard to the myeloid compartment, our experiments confirmed[12, 27] age-related increase in neutrophil granulocytes, as well as monocytes in the peripheral blood (Fig. 5t–v). EOD did not appear to attenuate the aging-associated

increases with regard to these myeloid cell populations in the peripheral blood (Fig. 5t–v).

**EOD did not alter aging-associated changes in plasma IgE.** We performed ELISAs to measure plasma IgE immunoglobulin levels, which are expected to increase with advanced age[12]. Our studies confirmed increased plasma IgE concentrations in aged mice (Fig. 5x). EOD did not ameliorate these aging-associated elevations in plasma IgE concentrations (Fig. 5x).

**EOD and age effects on clinical chemistry measures.** In line with previous observations[12, 28], we observed a number of clinical chemistry measures influenced by age (plasma taken from animals in the fed state). We observed aging-associated changes in plasma concentrations of sodium (increased in old mice), potassium (increased in old mice), urea (increased in old mice), glucose (decreased in old mice), albumin (decreased in old mice), unsaturated iron-binding capacity (increased in old mice), activity of alkaline phosphatase (increased in old mice), and lactate dehydrogenase activity (increased in old mice) (Fig. 6a–h;

full data set in Supplementary Table 16). There appeared to be changes with regard to plasma concentrations of fructosamine (decreased in old mice) and α-amylase activity (increased in old mice) but without reaching statistical significance (Supplementary Table 16). EOD had no measurable antagonistic effect on several

of these aging-associated changes, including those regarding plasma concentrations of sodium, potassium, glucose, albumin, and fructosamine. EOD did, however, appear to ameliorate aging-associated changes in plasma concentrations of urea, unsaturated iron-binding capacity, alkaline phosphatase activity, lactate

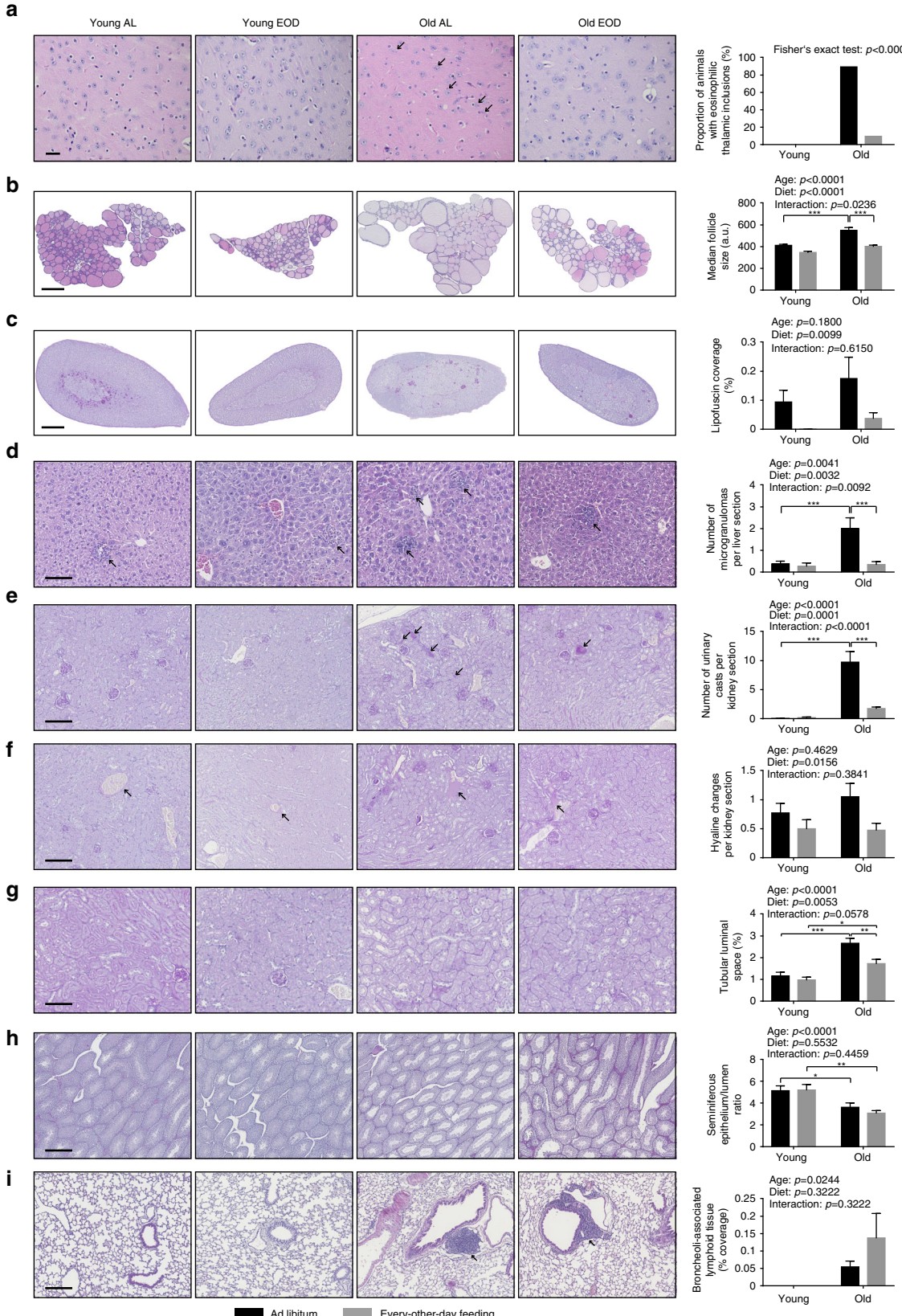

dehydrogenase activity, and α-amylase activity (Fig. 6c, f–h; Supplementary Table 16). Recovery of plasma urea concentration was the only parameter with EOD effects specifically in the aged cohort subjected to almost lifelong treatment, whereas all other parameters were influenced by EOD in similar ways in young mice on short-term EOD and aged mice on almost lifelong EOD.

We also noted additional main effects of EOD: EOD mice showed decreased plasma concentrations of calcium, phosphate, total protein, and triglycerides (Supplementary Table 16). Strongest effects were seen, unexpectedly, for parameters associated with iron metabolism: While total iron-binding capacity was only slightly decreased in EOD animals, we found a strong decrease of unsaturated iron-binding capacity and a corresponding increase of plasma iron levels in EOD mice (Fig. 6f and Supplementary Table 16). This difference was stronger in the young group than in aged animals.

A subset of parameters was measured in plasma isolated after a 6 h food withdrawal on a regular fasting day (full data set in Supplementary Table 17). After fasting, non-esterified fatty acids showed a significant main effect of age (decreased in old mice), without apparent correction by EOD (Fig. 6i). Several parameters, such as plasma levels of cholesterol, high-density lipoprotein cholesterol, triglycerides, and glucose, were decreased in EOD mice in an age-independent fashion (Supplementary Table 17).

**Glucose homeostasis in EOD mice.** To assess possible age- or diet-related changes in glucose tolerance, we performed an intraperitoneal glucose tolerance test (IpGTT) in young and old EOD and control animals (this was performed after 6 h of fasting on a day without access to food). We observed a significant main effect of EOD on basal glucose levels with levels reduced in EOD mice independent of age (Fig. 6k). Glucose tolerance was significantly improved in EOD mice of both age groups (Fig. 6k). Notably, inter-individual variance of glucose tolerance was much higher in aged control animals than in aged EOD mice. We did not observe significant main effects of age with regard to basal glucose levels and glucose tolerance.

We also measured insulin levels in young and old EOD and control animals (after 6 h of fasting on a day without food access). Consistent with published work[36], aged AL-fed mice showed elevated insulin levels compared to young controls (Tukey post hoc test following two-way ANOVA, old/AL vs. young/AL: $p = 0.0004$) (Fig. 6j). These analyses also revealed clear decreases in insulin levels in aged EOD mice (Tukey post hoc test following two-way ANOVA, old/AL vs. old/EOD: $p = 0.0005$) (Fig. 6j), likely reflecting increased insulin sensitivity in these animals.

**EOD ameliorated some aging-associated histological changes.** We used histopathology to assess young and old EOD and control animals for the presence of eosinophilic thalamic inclusions, cytoplasmic inclusions in thalamic neurons that progressively develop during aging in mice[37] and men[38]. Consistent with published data[37], eosinophilic thalamic inclusions were present in most of the AL-fed aged mice (Fig. 7a). These inclusions were rarely found in the old EOD animals (Fig. 7a). Eosinophilic thalamic inclusions were not present in young mice, either on EOD or AL access to food.

Aging-related changes in the thyroid gland include increasing follicle size and decreasing colloid resorption (reflecting decreasing thyroid activity in the aged animal)[39]. We performed morphometric histopathological analyses to assess follicle size distribution in the thyroid gland of young and old EOD and control animals. These studies revealed the expected[12, 39] aging-associated increase in thyroid follicle size (Fig. 7b). EOD reduced thyroid follicle size in both young mice on short-term EOD and aged mice subjected to almost lifelong EOD (Fig. 7b).

Aging-related histopathological alterations in the adrenal glands include adrenal adenomas, subcapsular cell hyperplasias, and lipofuscin depositions[40]. We employed morphometric histopathology to assess quantitatively the presence of lipofuscin deposits in the adrenal glands of young and old EOD mice and controls. Statistical analyses revealed an overall decrease in adrenal gland lipofuscin deposits in EOD mice (Fig. 7c).

Age-associated changes in the liver include fat accumulation, fibrotic changes, inflammatory/immune alterations, and cancers[12, 41]. We performed a histopathological assessment of liver-associated aging phenotypes in EOD mice and age-matched controls. AL-fed aged animals displayed microgranulomas (some with councilman bodies) and visible Kupffer cells (usually a sign for activation) (Fig. 7d). The number of hepatic microgranulomas was decreased in aged EOD mice (Tukey post hoc test following two-way ANOVA, old/AL vs. old/EOD: $p = 0.0004$) (Fig. 7d).

Aging-associated pathological changes in the kidney include glomerulosclerosis, loss of renal mass, and hyalinization of afferent arterioles[12, 42–44]. Obvious glomerulosclerotic changes were found in only one aged control animal. Quantitative assessment of urinary casts revealed that these features were largely absent in young mice but were prominent in aged AL animals and reduced in aged EOD mice (Tukey post hoc test following two-way ANOVA, old/AL vs. old/EOD: $p < 0.0001$) (Fig. 7e). Hyaline vascular changes were detected to some extent in both young and old mice, though these alterations were more pronounced in aged animals. EOD decreased hyaline vascular changes across age groups (Fig. 7f). With advanced age, tubular epithelial cells become atrophic, which is associated with an increase in tubular luminal space[42]. We used morphometric histopathology to quantify tubular luminal space on kidney sections. These analyses revealed a significant main effect of age on this measure, which was attenuated by EOD (Fig. 7g).

**EOD had limited effects on testicular atrophy and BALT.** Histopathological analyses of the male reproductive tract revealed the expected[12, 45] signs of testicular atrophy in aged mice. Consistent with age-dependent loss of testis weight (Supplementary Fig. 1n), seminiferous epithelium/lumen ratio of seminiferous tubules was significantly decreased in both old animal cohorts regardless of diet (Fig. 7h).

**Fig. 7** Histopathology. Representative images of sections derived from the **a** brain (scale bar = 20 μm), **b** thyroid gland (scale bar = 500 μm), **c** adrenal gland (scale bar = 500 μm), **d** liver (scale bar = 100 μm), **e–g** kidney (in **e**, **f**: scale bar = 200 μm; in **g**: scale bar = 100 μm), **h** testis (scale bar = 500 μm), and **i** lung (scale bar = 200 μm) are shown on the *left*. Quantification of **a** proportion of animals bearing eosinophilic thalamic inclusions (indicated by *arrows*), **b** median follicle size of the thyroid gland, **c** relative surface area covered by lipofuscin deposits within the adrenal gland, **d** number of hepatic microgranulomas (indicated by *arrows*), **e** number of urinary casts in the kidney (indicated by *arrows*), **f** number of hyaline changes in renal arterioles (indicated by *arrows*), **g** the relative surface area on renal sections covered by tubular luminal space, **h** ratio of testicular seminiferous epithelium/lumen in seminiferous tubules, and **i** coverage of broncheoli-associated lymphoid tissues (BALT) in the lung (indicated by *arrows*) are presented in panels on the *right*. Data in **b** are shown as median ± SEM and data in **c–i** are presented as mean ± SEM. Statistical analyses was performed using Fisher's exact test for **a** or two-way ANOVA for **b–i** (young + AL: $n = 16$, young + EOD: $n = 16$, old + AL: $n = 19$, old + EOD: $n = 20$). *AL* ad libitum, *EOD* every-other-day feeding

Aging is associated with the accumulation of lymphatic cells in various organs[12, 14], consistent with a proinflammatory state in aged individuals. We performed histopathological assessments to determine if aging had an effect on the presence of lymphatic tissue in the lung (bronchioli-associated lymphatic tissue, BALT) and, if so, whether EOD had an effect on this aging trait. BALT was present to various degrees in some of the aged mice irrespective of diet group (Fig. 7i), but not in young animals. EOD did not appear to inhibit the accumulation of BALT in the lung during aging.

**RNA-seq did not show EOD effects on transcriptional changes.** We employed RNA-seq analyses to determine whether EOD-modified transcriptional changes associated with aging (young/AL: $n = 4$ mice; young/EOD: $n = 3$ mice; old/AL: $n = 4$ mice; old/EOD: $n = 4$ mice; starting material: whole brain). Differential expression analyses identified 196 genes differentially expressed (FDR < 0.1) in old vs. young brains (Supplementary Data 1). Top differentially expressed genes included various immune regulators and effectors, such as *Lyz2*, *Il33*, and components of the complement system (e.g., *C4b*, *C3*, and *C1qa*). Pathway analyses comparing these 196 differentially expressed genes to the entire set of genes revealed many changes consistent with proinflammatory alterations in the aged brain (Supplementary Data 2). We did not detect genes differentially expressed as a function of diet (EOD vs. AL; FDR < 0.1) or with a significant (FDR < 0.1) interaction between the factors age and diet.

**Limited EOD effects on aging.** Among the 239 parameters tested, we detected 116 aging phenotypes (Fig. 1h). Surprisingly, in only seven cases of those, did we obtain evidence for a prevention of aging (Supplementary Table 2); examples are the prevention of eosinophilic thalamic inclusions in brain and urinary casts in the kidney (Fig. 7a, e). In 33 out of the 116 aging phenotypes, we observed similar EOD effects on age-sensitive parameters in young mice on short-term treatment and aged mice on almost lifelong treatment (Supplementary Table 3). Examples of traits falling in this category are aging-associated alteration in exploratory locomotor activity, visual acuity, HCT, cardiac dimensional measures, thyroid follicle size, T cell, and CD11c$^+$ NK cell frequencies (Figs. 2, 3, 5, and 7). The majority of the aging traits (76 out of the 116 parameters) examined in the present study was not measurably improved by EOD (Supplementary Table 4), including aging-associated changes in the optical density of the lens, a range of neurobehavioral functions, platelet counts, plasma immunoglobulin concentrations, and a number of clinical chemistry measures (e.g., levels of plasma sodium, potassium, or albumin) (Figs. 2, 5, and 6). Some age-sensitive parameters (14 parameters; Supplementary Table 4), such as decreased thickness of cortical bone and decreased responsiveness upon assessment of auditory function, were more pronounced in EOD mice than age-matched controls (Figs. 2 and 3).

In order to further understand the overall relationship of aging and EOD effects in young and old mice, we performed correlation analyses comparing the effect sizes of age and EOD on the parameters included in the present study. Comparison of effect sizes revealed an inverse correlation of aging effects and EOD effects (Fig. 1i). EOD effect sizes in young mice were tightly correlated with EOD effect sizes in aged mice (Fig. 1j). This was also the case when the analysis was restricted to aging traits ameliorated by EOD (Fig. 1j), indicating that EOD had overall similar effects in aged mice subjected to almost lifelong EOD and in young mice subjected to a short-term treatment.

## Discussion

Here we report on a large-scale assessment of DR effects, employing an EOD regimen, on aging and lifespan in male C57BL/6J mice. Consistent with previous EOD experiments[8, 9], as well as studies using other DR regimens, such as chronic CR[4–7], we observed an extension of median, as well as maximal lifespan in animals subjected to EOD. Murine lifespan extension under DR could in principle be due to isolated effects on specific life-limiting pathologies, in particular cancers, or could alternatively be caused by cancer suppression accompanied by broader anti-aging effects in mice. We show that longevity in male C57BL/6J mice subjected to EOD is associated with a delayed onset of lethal neoplastic disorders that typically limit natural lifespan in many mouse strains[6, 14]. Surprisingly, our large-scale analysis of >200 phenotypes across >20 tissues revealed only limited evidence for an EOD-induced delay of aging in mice.

DR regimens commonly used in rodents are EOD (employed in the present study) and CR. One of the important differences between these two protocols is the net calorie intake. Daily food supply of animals assigned to chronic CR is usually cut down to 60% of the amount consumed AL by age-matched controls (equivalent with 40% restriction). Chronic CR results in lifespan extension and pronounced growth retardation such that body weight (−42% in rat, −35% in mouse), fat mass (−70% in rat), and many organs weights (heart: −29% in rat; liver: −34% in rat, −31% in mouse; kidney: −33% in rat, −10% in mouse; spleen: −50% in rat, −66% in mouse; prostate: −25% in rat) are overall decreased in adulthood[6, 7]. Exceptions are sizes of brain and testis, which remain unaffected by chronic CR[7].

In contrast, animals subjected to EOD quickly adjust feeding to times of food availability and, therefore, show more modest reductions in net calorie intake compared to CR. In line with the CR findings mentioned above, EOD had strongest effects on organ weight with regard to the spleen (−31% in young, −56% in old). However, body length, fat mass, and the weights of most organs were affected more modestly by EOD. Thus, when compared to CR, EOD avoids partially the CR-associated growth retardation, while still affording the benefit of lifespan extension and other health benefits, such as improved insulin sensitivity and hippocampal neuroprotection against excitotoxic injury[3, 9].

The main causes of death in many mouse strains are neoplastic pathologies[6, 14]. Therefore, it is likely that a generally important aspect of lifespan-extending interventions in mice is the suppression of cancer development in the context of natural aging[46]. CR is known for its strong potential to suppress neoplasia[4, 5, 47]. EOD has been shown to exert protection in various tumor-prone rodent models[48, 49]. We here show that EOD reduces tumor burden and delays lethal neoplastic disease during natural aging in C57BL/6J mice. These considerations emphasize the need for studies complementing lifespan analyses with detailed analyses of treatment effects on a broad range of aging markers in different tissues and organs.

Prior EOD studies had examined restriction effects on a few additional parameters besides lifespan. Consistent with our observation of elevated spontaneous locomotor activity in EOD mice (in an open-field assay), previous experiments in rats had found increased levels of voluntary wheel running under EOD restriction[50]. EOD exerted cardioprotective effects in rodent ischemia models and, in line with the observations in the present study, influenced a number of cardiac dimensional measures consistent with a retardation of heart growth in restricted animals[51]. We also confirmed prior results regarding expected reductions of plasma glucose, triglyceride, and insulin concentrations in rodents subjected to EOD[52].

Under conditions of food shortage, nocturnal rodents, such as mice, are known to shift their activity pattern to also cover parts

of the light phase[53, 54]; this could represent an adaptive response that increases the chance to acquire food that is available only during restricted temporal windows[53]. Accordingly, altered circadian activity patterns have to be taken into account as potential confounds when considering outcomes of DR studies, including those employing EOD and CR[54, 55]. However, prior analyses, using CR, showed that feeding time influenced circadian rhythms, but did not affect CR-induced lifespan extension[56–58]. Moreover, AL-fed mice subjected to weekly 12-h shifts of their light-dark schedule during much of their lifetime did not show alterations in life expectancy compared to controls[57]. These data support the notion that DR effects on circadian rhythms are independent of longevity effects. Additional studies are needed to clarify whether DR-induced circadian alterations might interact with DR effects on aging phenotypes other than age-related mortality.

Our metabolic profiling experiments indicated that longevity in EOD mice was not associated with reduced, but rather slightly increased rates of energy expenditure, as evidenced by body weight-adjusted oxygen consumption, carbon dioxide emission, and heat production (HP). On the basis of the circadian patterns of food intake and RER values (though not necessarily overall locomotor activity), our metabolic analyses also provided some evidence for the expected alterations in circadian activity patterns in EOD mice (see above), which were particularly evident in the young cohort of animals. These changes were less clear in aged mice on almost lifelong EOD, raising the possibility that DR-associated shifts in circadian activity and associated metabolic parameters may be transient in nature and return to more typical nocturnal patterns with extended exposure to DR.

Our study had several limitations. The data were collected in one specific mouse strain (i.e., C57BL/6J) and sex (i.e., male). We focused our analyses on two specific age groups, including a young cohort with DR beginning at 8 weeks of age and phenotypic analyses commencing at ~3 months of age, as well as an aged cohort with DR commencing at 8 weeks of age and phenotyping starting at ca. 21 months of age. We did not examine additional intermediate or older cohorts. Future studies including such additional groups could define more detailed lifetime trajectories of the different traits examined. Furthermore, we would like to note that additional studies are needed to address whether other methods of DR, such as CR, have broader effects with regard to delaying natural aging.

Two important studies using rhesus macaques (Macaca mulatta) were launched in the late 1980s to explore CR-driven long-term survival and health outcomes in primates. The first study was initiated at the National Institute on Aging (NIA) in 1987, the latter started at the Wisconsin National Primate Research Center (WNPRC) in 1989. While CR extended lifespan in the WNPRC cohort, survival was not improved in the NIA study[59, 60]. This discrepancy in survival outcomes might be attributable to differences in study design, including feeding protocols (e.g., control monkeys in the WNPRC study ate AL, while controls in the NIA study received defined daily food portions)[61]. The incidence of specific age-related diseases (including cancer and diabetes) was reduced by CR in both the WNPRC and NIA study, showing that some of the expected health benefits are reproducible across studies. The impact of EOD in non-human primates has not been investigated so far and remains to be determined in future studies.

From a translational point of view, EOD or other forms of IF are attractive because of the overall low degree of food limitation associated with this form of DR[11]. The feasibility of applying IF to humans has been validated in a number of short-term trials over the past decade. Closely matching our findings in mice, IF decreased body weight and modulated parameters associated with insulin and fat metabolism after a few weeks of treatment[62].

However, these studies focused on acute effects, included only a limited set of measures and long-term health outcomes remain unexplored. Thus, the current study provides a foundation for the design of future long-term IF trials in humans.

In conclusion, our data indicate that EOD-induced lifespan extension in male C57BL/6J mice is sufficiently explained by a delay in lethal neoplastic disorders. Our large-scale assessment of molecular, cellular, physiological, and histopathological phenotypes in aging mice identified only a small proportion of aging traits delayed by EOD, suggesting that lifespan extension was not associated with a broad slowing of the aging process in EOD mice.

## Methods

**Mice.** For both, survival analysis as well as broad phenotyping of aging traits, male C57BL/6J mice were obtained as one cohort from Charles River Laboratories and were randomly assigned to the AL (AL = control) or every-other-day fasting (EOD) regimen. EOD was initiated at 2 months of age. All animals received a standard rodent chow (Altromin 1314) and were kept on their respective diet regimen for the entirety of the study (see below for more details regarding EOD).

Body weight was measured weekly over the entire course of the study. Food intake was estimated monthly by averaging food consumption on 4 consecutive days covering two fasting and two feeding days. For analyses of age- and EOD-sensitive traits, we analyzed a large set of molecular, cellular, physiological, and histopathological parameters in young and old EOD and control mice (age and number of animals at initiation of analyses: 21 months old control, $n = 24$ mice; 21 months old EOD, $n = 24$ mice; 3 months young control, $n = 16$ mice; 3 months young EOD, $n = 16$ mice). The following analyses were performed (in the order mentioned)[63]: open field (week 1), modified SHIRPA (week 1), grip strength (week 1), rotarod (week 2), acoustic startle response, and PPI (week 2), hot plate test (week 4), TEWL test (week 4), indirect calorimetry (week 5), NMR-based body composition analysis (week 6), glucose tolerance test (week 7), awake electrocardiography, and echocardiography (week 8), Scheimpflug imaging (week 9), optical coherence tomography (week 9), laser interference biometry (week 9), virtual drum vision test (week 9), clinical chemistry (week 10), hematology (week 10), FACS analysis of peripheral blood leukocytes (week 10), Bioplex ELISA (Ig concentrations) (week 10), auditory brain stem response (week 11), X-ray/bone densitometry (week 11), and pathology (week 13). If not stated otherwise, we commenced with data collection in the morning after a feeding day to avoid artificial effects driven by hunger. Mice (in their home cages) were habituated to the test room for a period of 15 min prior to commencement of analyses for most experimental procedures (unless stated otherwise). In all cases, the experimenter was blinded to the animal group assignments.

Mice were housed under specific pathogen-free conditions, i.e., they were free of defined murine pathogens according to FELASA guidelines. The animals were housed in groups of four mice per cage at initiation of the study. Animals were kept at constant temperature of 22 °C on a 12:12 h light/dark cycle. Mice received water AL. Local and federal regulations regarding animal welfare were followed. The present study was approved by "Landesamt für Natur, Umwelt und Verbraucherschutz Nordrhein-Westfalen" (Recklinghausen, Germany), as well as "Regierung von Oberbayern" (Munich, Germany).

**EOD Regimen.** We chose a DR regimen (i.e., EOD) previously shown to extend lifespan in mice[9] and compatible with group housing the animals. Unlike other DR paradigms (such as restriction of calorie intake to 60% of what animals would eat AL), the feeding situation in the EOD paradigm is not associated with competition for food (because food is available in excess during the feeding day) and therefore does not require single housing of the animals.

Animals fed AL were granted unlimited access to food anytime. EOD was conducted according to a published protocol allowing free access to food for 24 h alternated by food deprivation for 24 h (also referred to as EOD)[9]. Food was provided to the EOD cohorts at 9 am and withdrawn at 9 a.m. on the next morning as described previously[9]. EOD was initiated at 2 months of age, corresponding to the age of EOD onset associated with the largest effect size on lifespan in a previous study on male C57BL/6J mice[9]. All animals received a standard rodent chow (Altromin 1314; closely matching the composition of the rodent chow used in a previous EOD longevity study on male C57BL/6J mice[9]) and were kept on their respective diet regimen for the entirety of the study. The Altromin 1314 chow came in solid pellets. Careful pilot analyses showed that mice did not crumble these pellets. Accordingly, removing the pellets on the restriction days was sufficient to fully deprive the animals of food (no cage change required).

**Open-field.** For the open-field test, an apparatus (ActiMot, TSE) consisting of a transparent and infrared light-permeable acrylic test box (internal measurements: 45.5 cm × 45.5 cm × 39.5 cm), equipped with two pairs of light beam strips, was used. The center of the test arena was illuminated with about 200 lux and the corners were enlightened by ~150 lux. Animals were transported to an area directly

adjacent to the testing room 30 min prior to open-field analyses. The 20-min open-field test itself was then started right away given that the novelty aspect of the environment is a crucial component of the test. Parameters analyzed were: total distance traveled, number of rears, total resting time, average speed, distance traveled in periphery, speed in periphery, resting time in periphery, permanence in periphery, distance traveled in center zone, time spent in center zone, speed in center zone, resting time in center zone, permanence in center zone, latency to enter the center zone, and number of center zone entries.

**Acoustic startle response and PPI**. Animals were transported to an area directly adjacent to the testing room 30 min prior to behavioral analyses. Next, we employed a 5-min acclimation period to the test compartment (i.e., a mouse restrainer), before the actual testing began. The PPI protocol was based on the Eumorphia protocol. In brief, PPI was assessed using a startle apparatus set-up (Med Associates), including four identical sound-attenuating cubicles. Background noise was 65 dB, and startle pulses were bursts of white noise (40 ms). A session was initiated with a 5-min acclimation period followed by five presentations of leader startle pulses (110 dB) that were excluded from statistical analysis. Trial types for the PPI included four different pre-pulse intensities (67, 69, 73, and 81 dB); each pre-pulse preceded the startle pulse (110 dB) by a 50 ms inter-stimulus interval. Each trial type was presented 10 times in random order, organized in 10 blocks, each trial type occurring once per block. Inter-trial intervals varied from 20–30 s.

**Modified SHIRPA**. The modified SHIRPA screen was proposed as a rapid, comprehensive, and semi-quantitative screening method for qualitative analysis of abnormal phenotypes in mouse strains[64]. We examined the mice using 17 test parameters to detect phenotypic differences between dietary restricted and control aged mice, as well as young controls. Each test parameter contributes to an overall assessment of muscle, lower motor neuron, spinocerebellar, sensory and autonomic function and is scored qualitatively following a defined rating scale. Assessment of each animal began with the observation of undisturbed behavior in a glass cylinder (11 cm in diameter). The mice were then transferred to an arena consisting of a clear Perspex box (420 mm × 260 mm × 180 mm) in which a Perspex sheet on the floor is marked with 15 squares. Locomotor activity and motor behavior within this area was observed. This was followed by a sequence of manipulations testing reflexes. Throughout the entire procedure, abnormal behavior, biting, defecation, and vocalization were recorded. Between testing of each mouse, fecal pellets and urination were removed from the viewing jar and arena. The following functions were evaluated in the context of the modified SHIRPA screen: muscle/lower motor neuron function (body position, gait, tail elevation, defecation, and urination), spinocerebellar function (body position, gait, tail elevation, and limp grasping), sensory function (transfer arousal, touch escape, gait, pinna reflex, and trunk curl), autonomic function (defecation and urination), neurological reflexes (contact righting reflex and pinna reflex), general appearance (body weight, body position, transfer arousal, touch escape, vocalization, biting, spontaneous activity, locomotor activity, and abnormal behavior).

**Grip strength**. A grip strength meter system (Bioseb) was used to determine grip strength of the mice. The task exploits the tendency of the mice to grasp a horizontal metal grid while being pulled at the tail. During the set-up for the trial, the mice grasped a special adjustable grid mounted on a force sensor. Mice were then allowed to hold on to the grid with either two or four paws. Three trials over the course of 1 min were conducted for each mouse. Mean values were used to represent the grip strength of a mouse.

**Rotarod**. This task was carried out to determine balance, motor coordination, and motor learning abilities on an accelerating rotarod (Bioseb). After the mice had been placed on the test apparatus, rod rotations were continuously accelerated from 4 to 40 rpm within the 5-min trial period. Trials ended when animals fell of the rod, showed passive cycling or when 5 min expired, whichever came first. For each animal, three trials with inter-trial intervals of 15 min were performed. Data are presented as the average of the three trials.

**Auditory brain stem response**. The auditory brain stem response test was performed to assess hearing sensitivity in a non-invasive manner. Anesthetized mice (ketamine/xylazine-based anesthesia) were transferred onto a heating blanket in the acoustic chamber, followed by the positioning of three subcutaneous needle electrodes. For threshold determination, we used clicks (0.01 ms duration) or tone pips (6, 12, 18, 24, and 30 kHz of 5 ms duration, 1 ms rise/fall time) over a range of intensity levels (from 5 to 85 dB SPL in 5 dB steps). The sound intensity threshold was determined manually based on the first appearance of the characteristic waveform.

**Hot plate test**. The hot plate test was performed as described elsewhere[12]. The animals were placed on a pre-heated metal surface ($52 \pm 0.2$ °C) which was surrounded by a circular Plexiglas wall (20 cm high, 28 cm in diameter) in order to curtail the animals' locomotion (Hot plate system, TSE). Mice were removed from the plate after one of three responses indicative of a painful sensation occurred: hind paw licking, hind paw shake/flutter or jumping. The latency was recorded to the nearest 0.1 s. The maximum testing duration was limited to 30 s to avoid tissue injury.

**Indirect calorimetry**. Metabolic assessments were performed at room temperature (23 °C) with a 12:12 h light/dark cycle in the room (lights on 6 a.m. CET, lights off 6 p.m. CET) for 2 consecutive days. The indirect calorimetry trial was conducted in home cages equipped with food hoppers and water bottles. Mice were placed individually into these respirometry cages for about 2 h before the actual recordings were started. We carefully monitored that mice successfully used food hoppers and water bottles during this habituation period and also throughout the consecutive 47 h. AL animals were granted free access to food and water, whereas the EOD cohorts were only fed during the first 24 h. Wood shavings and paper tissue was provided as bedding material. Metabolic cages were set up in a ventilated cabinet continuously supplied with fresh air. Variations in $O_2$ and $CO_2$ levels were recorded by high precision $CO_2$ and $O_2$ sensors in each individual cage. Combined with parallel airflow measurements, this enabled calculation of oxygen consumption (expressed as ml $O_2$/h/animal) over a given time period. The system also monitored $CO_2$ production, and, hence, it was possible to determine the RER and heat production (HP). The RER was calculated as the ratio $VCO_2/VO_2$. HP was calculated from $VO_2$ and RER using the formula: HP (mW) = (4.44 + 1.43 × RER) × $VO_2$ (ml/h). In addition, body mass was recorded before and after gas exchange measurements. Cumulative food intake and water consumption were monitored continuously. Physical activity (distance traveled, number of rearings) was measured using infrared light beam frames set-up around the cages.

**Body composition analysis**. Body composition (lean tissue, body fat) was determined using a body composition analyzer (Bruker MiniSpec) based on TD-NMR.

**Echocardiography**. Transthoracic echocardiography was performed using a Vevo 2100 Imaging System (Visual Sonics) with a 30 MHz probe to assess left ventricular function. Examinations were performed on conscious mice to avoid effects of anesthesia on cardiac function. Left ventricular parasternal short- and long-axis views were obtained in B-mode imaging and left ventricular parasternal short-axis views were obtained in M-mode imaging at the papillary muscle level. According to the American Society of Echocardiography leading edge method[65], LVIDd, left ventricular end-systolic internal diameter (LVIDs), diastolic and systolic septal wall thickness (IVSd and IVSs) and left ventricular diastolic and systolic posterior wall thickness (LVPWd and LVPWs) were determined using short-axis M-mode images derived from three consecutive beats. Fractional shortening (FS) was calculated as FS% = ((LVIDd – LVIDs)/LVIDd) × 100. Ejection fraction (EF) was calculated as EF% = 100 × ((LVvolD – LVvolS) LVvolD) with LVvol = ((7.0/(2.4 + LVID) × LVID$^3$). The LV mass corr was calculated as LV mass corr = 0.8 (1.053 × ((LVIDd + LVPWd + IVSd)$^3$ – LVIDd$^3$)). The stroke volume (SV) was defined as the volume of blood ejected by one ventricle during one beat. SV of the left ventricle was obtained by substracting end-systolic volume from end-diastolic volume. The cardiac output was defined as the volume of blood pumped by the heart per minute. In addition, heart rate (HR) and respiratory rate were calculated by measuring three systolic intervals and three respiratory intervals, respectively.

**Electrocardiography**. Electrocardiographs (ECGs) were recorded with the ECGenie (Mouse Specifity) using conscious mice and were analyzed via e-Mouse software (Mouse Specifics). Cardiac electrical activity was detected non-invasively through the animals' paws. For each animal, intervals and amplitudes were assessed from continuous recordings of at least 15 ECG signals. Peak detections were used to calculate the HR. HR variability (HRV) was calculated as the mean of the differences between sequential heart beats. Interpretations of P, Q, R, S, and T for each beat were plotted so that unfiltered noise or motion artifacts were rejected. Calculations of the mean of the ECG time intervals for each set of waveforms followed. The corrected QT interval (QTc) was calculated by dividing the QT interval by the square root of the preceding RR interval. QT dispersion was measured as inter-lead variability of QT intervals. The QTc dispersion was calculated as the rate QTc dispersion. The mean SR amplitude was calculated as the mean amplitude of signal measured from the signal minimum line to peak of QRS. The mean R amplitude was calculated as the mean amplitude of signal measured from isoelectric line to peak of QRS. rMSSD was calculated as root mean square of the successive differences of neighboring RR intervals. pNN50 was calculated as percentage of adjacent RR intervals with differences larger than 50 ms. Parameters analyzed were: HR, HRV, signal intensity, RR interval, PQ interval, PR interval, QRS interval, QT interval, ST interval, QTc, QT dispersion, QTc dispersion, SR amplitude, R amplitude, rMMSD, and pNN50.

**Optical coherence tomography**. The posterior part of the eye (fundus and retina) was evaluated by a Spectralis OCT (Heidelberg Engineering) equipped with a 78 diopter double aspheric lens (Volk Optical) that was directly mounted to the outlet of the apparatus. A contact lens with a focal length of 10 mm (Roland Consult) was connected to the eye of the mouse with a drop of methylcellulose (Methocel 2%, OmniVision). Prior of testing, 1% atropine was applied to dilate pupils. For

measurements, anesthetized mice were placed on a platform in front of the Spectralis OCT such that the eye was directly facing the lens of the recording unit. Images were taken as described previously[66]. Retinal thickness was calculated with the thickness profile tool provided. Parameters evaluated were: pigmentation (fundus), morphology of optic disk (fundus), number of main blood vessels (fundus), retinal thickness, and morphology of retinal layers.

**Scheimpflug imaging**. The Pentacam digital camera system (Oculus) was used to examine properties of cornea and lens. Mice were put on a platform such that the center of the eye faced the vertical light slit (light source: LEDs, 475 nm). Assisted by the software, optimal focus was obtained by distance adjustments between eye and camera[67]. Lens densities (minimum, mean, maximum) were assessed quantitatively, whereas the morphologies of cornea and lens were evaluated qualitatively.

**Virtual drum vision test**. Vision tests were performed using a virtual optomotor system (Cerebral Mechanics)[68]. Mice were placed on an elevated platform in the center of an area surrounded by four monitors. A cylinder comprised with a sine wave grating was drawn in virtual three-dimensional space and rotated around the animal. During testing, the behavior of the animal was tracked by a camera mounted above. Lack of tracking the sine wave grating with reflexive head and neck movements was indicative for an inability to resolve the visual pattern. A randomized simple staircase test was used to quantify vision thresholds. Rotation speed and contrast were set to 12.0 d/s and 100%, respectively.

**Glucose tolerance test**. The IpGTT was performed after 6 h of food withdrawal on a fasting day. During the experiment, mice were kept separately in empty cages without food or bedding. Body mass of mice was determined before food removal and before the test was started. To determine the basal blood glucose level in the unfed state, a small drop of blood obtained by cutting the tip of the tail was analyzed with the Accu-Chek Aviva glucose analyzer (Roche). Subsequently, 2 g glucose/kg body mass was injected intraperitoneally using a sterile 20% glucose solution (B. Braun). We measured blood glucose levels in samples collected at several time points after injection (15, 30, 60, and 120 min). After completion of the experiment, mice were placed in their home cage with water and food available AL.

**Blood collection**. Mice were anesthetized with isoflurane and their retro-bulbar sinus was punctured with non-heparinized glass capillaries (1.0 mm in diameter; Neolab) to collect blood samples. For measurements after fasting, blood samples were taken after a 6 h food withdrawal on a fasting day and collected in heparinized sample tubes (Li-heparin, #078028, KABE). For measurements in the fed state, blood samples collected from fed mice were divided into two portions. The major portion was collected in a heparinized tube (Li-heparin, KABE). The smaller portion was collected (using the same capillary) in an EDTA-coated tube (KABE, #078035). Each tube was immediately inverted five times to achieve a homogeneous distribution of the anticoagulant. Li-heparin-coated tubes were stored at room temperature for 1-2 h. Afterwards, cells and plasma were separated by a centrifugation step (10 min, $5000 \times g$, 8 °C). Plasma was used for the immunology experiments (30 μl; immunoglobulin concentrations) and the clinical chemistry assessment (110 μl). The cell pellet was used for FACS-based immunology screen (i.e., quantitative assessment of peripheral blood leukocyte populations). Plasma samples for the clinical chemistry analyses were transferred into an Eppendorf tube and either used directly (plasma of unfed mice) or diluted 1:2 with distilled water (fed mice). The solution was mixed for a few seconds to prevent clotting and was then centrifuged again (10 min, $5000 \times g$, 8 °C). The EDTA blood samples were used for hematological analyses and kept on a rotary agitator at room temperature prior to analysis.

**Clinical chemistry**. Clinical chemistry parameters were assessed using a Beckman Coulter AU 480 auto-analyzer with few exceptions. Concentrations of free fatty acids were measured using a kit from Wako Chemicals GmbH (NEFA-HR2, Wako Chemicals) and levels of glycerol were analyzed employing a kit from Randox Laboratories. A broad set of parameters was measured including various enzyme activities, as well as plasma concentrations of specific substrates and electrolytes in fed mice (alkaline phosphatase (ALP); EC 3.1.3.1), α-amylase (EC 3.2.1.1), aspartate-aminotransferase (ASAT/GOT; EC 2.6.1.1), alanine-aminotransferase (ALAT/GPT; EC 2.6.1.2), lactate dehydrogenase (LDH; EC 1.1.1.27), total protein, albumin, glucose, lactate, fructosamine, cholesterol, triglycerides, urea, potassium, sodium, chloride, calcium, inorganic phosphate, iron, and unsaturated iron-binding capacity). For determination of clinical chemistry parameters in the fed state, blood was taken from all animals in the morning of a regular fasting day immediately before food withdrawal (i.e., both EOD and control mice had free access to food prior to blood collection). Another set of parameters was determined in samples derived from mice after a 6-h food withdrawal on a fasting day (i.e., both EOD and control mice were food deprived for 6 h prior to blood collection): Cholesterol, triglycerides, glucose, glycerol, non-esterified fatty acids (NEFA), high-density lipoprotein (HDL) cholesterol. Plasma concentration of non-HDL cholesterol was calculated. In addition, plasma insulin levels were determined in these samples by conventional ELISA (Mercodia ultrasensitive mouse insulin ELISA, Mercodia).

**Hematology**. A measure of 50 μl EDTA blood was diluted 1:5 in 200 μl of Sysmex Cell-Pack buffer prefilled tubes (Sysmex, Art.No. 99940020) and was used to determine complete blood cell counts using Sysmex XT2000iV. This device is a hematology analyzer combining electrical impedance technique, fluorescence staining, and flow cytometry to determine number and size of the different blood cells and to distinguish erythrocytes and reticulocytes as well as different types of leukocytes. MCV, mean platelet volume (MPV), PDW, and RDW were calculated directly from the cell volume measurements. HCT was determined by multiplying the MCV with the RBC count. Mean corpuscular hemoglobin (MCH) and mean corpuscular hemoglobin concentrations (MCHC) were calculated by dividing hemoglobin/RBC count (MCH) and hemoglobin/HCT (MCHC), respectively. Platelet large cell ratio (PLCR) was defined as the proportion of large platelets (>12 fl). PCT was calculated by multiplying the MPV with the platelet count. Parameters analyzed were: number of white blood cells (WBC), number of RBC, number of platelets (PLT), hemoglobin concentration (HGB), HCT, MCV, MCH,MCHC, RDW, MPV, PDW, PLCR (>12 fl), and PCT.

**FACS-based analysis of peripheral blood leukocytes (PBLs)**. PBLs were profiled from 20 μl whole blood per mouse. Each whole-blood sample was incubated at 4–10 °C for 5 min with Fc block (clone 2.4G2). The antibody mix was added afterwards and incubated at 4–10 °C for 40–60 min. Next, erythrocytes were lysed and a formalin-based fixation step was performed (BD FACS Lysing solution, Becton Dickinson). Finally, after washing with FACS staining buffer (PBS, 0.5 % BSA, 0.02 % sodium azide, pH 7.45), samples were analyzed on a 96-well plate using a HyperCyt sampler (IntelliCyt) and a ten-color flow cytometer (Gallios, Beckman Coulter). The acquisition threshold was set on the CD45-channel. A total number of 10,000–50,000 leukocytes per sample was examined. A software-based analysis (Flowjo, TreeStar) was used to quantify individual PBL frequencies. Gates for each parameter were based on "fluorescence minus one" controls[69]. FACS gating strategies are described in Supplementary Figs. 2–4.

The following main leukocyte populations were examined: B1 cells (CD5$^+$, CD19$^+$B220$^+$), B2 cells (CD5$^-$, CD19$^+$B220$^+$), T cells (CD3$^+$CD5$^+$), granulocytes (CD11b$^+$Ly6G$^+$), NK cells (CD3$^-$CD5$^-$, NKp46$^+$, and/or NK1.1$^+$), NKT cells (CD3$^+$CD5$^+$, NKp46$^+$, and/or NK1.1$^+$), monocytes (non-lymphocytes, non-granulocytes, and CD11b$^+$). The following monocyte subpopulations were analyzed: CD11c (positive or negative), Ly6C (high, medium, and negative). The T-cell population (T cells are defined as CD3$^+$CD5$^+$ and NKp46$^-$NK1.1$^-$) was subdivided based on the expression of Ly6C. B-cell (CD19$^+$B220$^+$) populations were subdivided according to the expression of CD11b and Ly6C. NK cell populations were subdivided based on the expression of CD11b, CD11c, and Ly6C.

**Immunoglobulin concentrations**. The plasma level of IgE was analyzed using a classical immunoassay isotype-specific sandwich ELISA (BD Pharmingen).

**TEWL**. TEWL measurements were performed in order to assess the quantity of water lost via diffusion and evaporation to the surrounding atmosphere. Mouse skin was assessed non-invasively with a special Tewameter (AquaFlux AF200) placed on the skin. TEWL (g/(m$^2$h)) was recorded over a short time period of 60-90 s.

**Micro-CT imaging and analysis**. Tibiae from young and aged mice were imaged using a SkyScan 1172 micro-CT scanner (Bruker) at 7.88 μm pixel resolution using 80 kV voltage, 100 mA current, and a 0.5 mm aluminum filter. The resulting projection images were reconstructed with SkyScan's NRECON package using a uniform attenuation coefficient, and analyzed using CTAn v.1.15. Cortical bone architecture in the distal tibia (197 μm from 25 slices halfway between the tibia-fibula junction and the distal end of the tibia) was evaluated, and relevant bone parameters (total tissue area, bone area, marrow area, cortical thickness, polar moment of inertia) were selected for further data analysis.

**Pathology**. After being killed via $CO_2$, mice were analyzed macroscopically and weighed (http://eulep.pdn.cam.ac.uk/Necropsy_of_the_Mouse/index.php). All organs (skin, heart, muscle, lung, brain, cerebellum, thymus, spleen, cervical lymph nodes, thyroid, parathyroid, adrenal gland, stomach, intestine, liver, pancreas, kidney, reproductive organs, and urinary bladder) were taken, fixed in 4% buffered formalin and embedded in paraffin for later histological examination. Two-μm-thick sections from each organ sample were generated and stained with hematoxylin and eosin, Periodic acid Schiff stain, Elastica van Gieson (EvG, Weigert's stain), and/or Movat pentachrome. Thereafter, all sections were scanned using a virtual slide system (NanozoomerHT2.0; Hamamatsu) and evaluated by a board-certified pathologist. In addition, specialized automated image segmentation software (Definiens Tissue Studio (Definiens) and CellProfiler[70]) was used to quantify age-related tissue changes via computer-assisted analyses.

**RNA sequencing**. Illumina next-generation sequencing libraries, generated from high-quality input RNAs, were analyzed on an Illumina HiSeq2000 system (Illumina) via 50 bp single end sequencing. Reads were quality and adapter trimmed with custom software. The resulting fastq files were aligned with STAR against the Ensembl GRCm38 reference genome. featureCounts was used on the resulting BAM files to generate unique counts. Those results were then input into DESeq2 for further analysis. Ingenuity pathway analysis (Ingenuity Systems) was employed for pathway analysis of differential expressed genes (FDR < 0.1). fastq and featureCounts files were deposited on NCBI's gene expression omnibus (GEO) under the accession code GSE96644.

**Statistics**. Unless stated otherwise, the data are presented as mean ± SEM and were analyzed by two-way ANOVAs with the between-subjects factors of age (old vs. young) and DR (EOD vs. AL). Tukey post hoc tests were applied if appropriate. Statistical analyses were carried out using GraphPad (version 6.07) and SPSS (version 23). $p < 0.05$ was considered statistically significant. $*p < 0.05$, $**p < 0.01$, and $***p < 0.001$.

**Data availability**. RNA sequencing data that support the findings of this study have been deposited in GEO under accession code GSE96644. All other data that support the findings of the current study are included within the article or the associated supplementary material.

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

## Acknowledgements

We thank the DZNE animal caretaker team, the DZNE animal facility management, the GMC technicians, as well as the GMC animal caretaker team for expert technical assistance and valuable support. The study was funded by the German Center for Neurodegenerative Diseases (DZNE), the German Federal Ministry of Education and Research (Infrafrontier Grant 01KX1012), the German Center for Diabetes Research (DZD), the German Center for Vertigo and Balance Disorders (Grant 01EO 0901) and by the Helmholtz Alliance for Mental Health in Ageing Society (HA-215).

## Author contributions

D.E. conceived and initiated the study; K.X., H.F., and D.E. planned and prepared the study; K.X., F.N., J.R., J.A.A.-P., O.V.A., L.B., R.B., L.G., K.S.H., S.M.H., D.J., I.L., K.M., B.L.P., I.R., B.R., D.P.R., S.S., I.T., and D.E. performed the experiments; K.X., F.N., J.R., J.A.A.-P., O.V.A., L.B., R.B., L.G., K.S.H., S.M.H., D.J., I.L., K.M., B.P., I.R., B.R., D.P.R., S.S., I.T., and D.E. analyzed the data; R.B., D.H.B., J.G., G.E., M.K., T.K., M.O., C.S.-W., M.W., E.W., W.W., A.Z., V.G.-D., H.F., M.H.A., and D.E. provided oversight and resources; D.E. and K.X. wrote the manuscript.

## Additional information

**Competing interests:** The authors declare no competing financial interests.

Kan Xie[1], Frauke Neff[2,3], Astrid Markert[1], Jan Rozman [3,4], Juan Antonio Aguilar-Pimentel[3,5], Oana Veronica Amarie[3,6], Lore Becker[3], Robert Brommage[3], Lillian Garrett[3,6], Kristin S. Henzel[1], Sabine M. Hölter [3,6], Dirk Janik[2,3], Isabelle Lehmann[1], Kristin Moreth [3], Brandon L. Pearson [1], Ildiko Racz[3,7], Birgit Rathkolb[3,4,8], Devon P. Ryan[1], Susanne Schröder[1], Irina Treise[3,9], Raffi Bekeredjian[10], Dirk H. Busch[9], Jochen Graw[6], Gerhard Ehninger[11], Martin Klingenspor[12], Thomas Klopstock [13,14,15,16], Markus Ollert[17,18], Michael Sandholzer[3,19], Carsten Schmidt-Weber[20,21], Marco Weiergräber[22], Eckhard Wolf[8], Wolfgang Wurst[6,15,16,23], Andreas Zimmer[7], Valerie Gailus-Durner[3], Helmut Fuchs[3], Martin Hrabě de Angelis [3,4,24] & Dan Ehninger[1]

[1]DZNE, German Center for Neurodegenerative Diseases, Ludwig-Erhard-Allee 2, 53175 Bonn, Germany. [2]Institute of Pathology, Helmholtz Zentrum München, German Research Center for Environmental Health, Ingolstädter Landstraße 1, 85764 Neuherberg, Germany. [3]German Mouse Clinic, Institute of Experimental Genetics, Helmholtz Zentrum München, German Research Center for Environmental Health, Ingolstädter Landstraße 1, 85764 Neuherberg, Germany. [4]Member of German Center for Diabetes Research (DZD), Ingolstädter Landstraße 1, 85764 München-Neuherberg, Germany. [5]Division of Environmental Dermatology and Allergy, Technische Universität München/Helmholtz Zentrum München, Ingolstädter Landstraße 1, 85764 Neuherberg, Germany. [6]Institute of Developmental Genetics, Helmholtz Zentrum München, German Research Center for Environmental Health, Ingolstädter Landstraße 1, 85764 Neuherberg, Germany. [7]Institute of Molecular Psychiatry, Medical Faculty, University of Bonn, Sigmund-Freud-Straße 25, 53105 Bonn, Germany. [8]Chair of Molecular Animal Breeding and Biotechnology, Gene Center, Ludwig-Maximilians-Universität München, Feodor Lynen-Straße 25, 81377 Munich, Germany. [9]Institute for Medical Microbiology, Immunology, and

Hygiene, Trogerstraße 30, Technische Universität München, 81675 Munich, Germany. [10]Department of Medicine III, Division of Cardiology, University of Heidelberg, Im Neuenheimer Feld 410, 69120 Heidelberg, Germany. [11]Department of Internal Medicine I, University Hospital Carl Gustav Carus, Technical University Dresden, Fetscherstraße 74, 01307 Dresden, Germany. [12]Molecular Nutritional Medicine, Else Kröner-Fresenius Center, Technische Universität München, Gregor-Mendel-Straße 2, 85350 Freising-Weihenstephan, Germany. [13]Friedrich-Baur-Institut, Department of Neurology, Ludwig-Maximilians-Universität München, Ziemssenstraße 1a, 80336 Munich, Germany. [14]German Center for Vertigo and Balance Disorders, University Hospital Munich, Campus Grosshadern, Marchioninistraße 15, 81377 Munich, Germany. [15]DZNE, German Center for Neurodegenerative Diseases, Schillerstraße 44, 80336 Munich, Germany. [16]Munich Cluster for Systems Neurology (SyNergy), 80336 Munich, Germany. [17]Department of Infection and Immunity, Luxembourg Institute of Health, Esch-sur-Alzette, Luxembourg. [18]Department of Dermatology and Allergy Center, Odense Research Center for Anaphylaxis, University of Southern Denmark, Odense, Denmark. [19]Comprehensive Pneumology Center, Helmholtz Zentrum München, German Research Center for Environmental Health, Ingolstädter Landstraße 1, 85764 Neuherberg, Germany. [20]Center of Allergy & Environment (ZAUM), Technische Universität München and Helmholtz Zentrum München, Biedersteiner Str. 29, 80802 Munich, Germany. [21]Member of the German Center for Lung Research (DZL), Aulweg 130, 35392 Gießen, Germany. [22]Research Group Experimental Neuropsychopharmacology, Federal Institute for Drugs and Medical Devices, Kurt-Georg-Kiesinger-Allee 3, 53175 Bonn, Germany. [23]Chair of Developmental Genetics, Technische Universität München, c/o Helmholtz Zentrum München, German Research Center for Environmental Health, Ingolstädter Landstraße 1, 85764 Neuherberg, Germany. [24]Chair of Experimental Genetics, Center of Life and Food Sciences Weihenstephan, Technische Universität München, 85350 Freising-Weihenstephan, Germany. Martin Hrabě de Angelis and Dan Ehninger contributed equally to this work.

