## [Peer Review File · Nature Communications]

Reviewers' Comments:

Reviewer #1 (Remarks to the Author)

The manuscript describes a dietary restriction regimen introduced on male C57B6 mice and its effects on different physiological outcomes. This is a large body of work done and most of the results are well presented. The experimental methods and the outcomes will also serve as a reference point for comparable studies. The authors should address the following points to strengthen the manuscript and to accurately represent the findings. Specifically, the authors must change the term IF to EOD for any publication in any journal. Similarly, the title is very misleading as it does not clearly articulate what would the authors expect if an intervention reduces aging (except making the mice immortal). The experimental method clearly introduces two variables – prolonged fasting between periods of food access, and significant perturbation of normal sleep-wake cycle. One of them is beneficial, while the chronic perturbation of sleep wake cycle has adverse effect on health. In this context, it can be argued that the adverse effects of sleep-wake perturbation can be reduced by the fasting period that is integral to EOD. At the same time, it can also be argued that the maximum potential benefits of fasting may be reduced by the sleep-wake disturbances that are intrinsic to the protocol. This becomes an important confound in extending the protocol to a different diet composition that mimic the human diet preferences of moderately high fat or high sucrose content. This intrinsic confound in the study is a HUGE factor in the accurate interpretation and future extension of study findings. While it is impractical to repeat any of the experiments, the authors must put some time into re-writing several parts of the manuscript to highlight and discuss this point.

Suggested new title "Every other day food intake extends murine lifespan, but has variable effects on markers of aging"

Major concern. The title of the manuscript should accurately reflect the procedure used. With different dietary restrictions, the term intermittent fasting is loosely used to cover many types of fasting regimens. The range of fasting regimens differs in the duration of fasting, caloric intake and have diverse significance for reduction to practice in humans. The authors must clearly state they tried Every Other Day feeding (EOD) in the abstract and change to title to reflect it is an EOD protocol. All figure panels, tables and in text reference to "intermittent fasting" must be replaced with EOD to reflect the factually correct procedure used in the manuscript.

The fasting method refers to a protocol used more than a quarter century old when researchers did not pay any attention to the natural eating and sleeping pattern of mice. Providing food to nocturnal mice at 9am local time forces them to stay awake and consume the food, which can perturb their natural cycle of feeding-fasting and sleep-wakefulness. This disruption is clearly evident in supplementary figure 4c. Essentially the EOD forces the mice to be awake and feed for almost 24 h on the day of feeding. Therefore, EOD is a combination of both 24 h fast every other day and a shiftwork like regimen. I strongly suggest the author place Figure 4c in the main figure to show the daily metabolic pattern in the cohorts. This 47 h RER is one of the important physiological phenotypes in which these cohorts differed and it should be clearly shown in the main text. With numerous papers in both animal and human models showing this diurnal perturbation can have adverse metabolic and neoplastic consequences, the conclusion of the manuscript does include a large confound. Therefore, it is inaccurate to conclude that the procedure had limited impact on aging.

The manuscript is presented with no clear focus on why some of the phenotypes were measured. It sounds like a list of capabilities the PIs have access to. For each phenotype measured, the author should include some information on how that specific phenotype changes with age, whether there is any evidence that any prior intervention has improved that specific age-dependent change in the phenotype and then make the interpretation whether this EOD intervention was effective or not in reducing the age-dependent change. This can be presented as an additional column in some of the supplementary tables. Without such clear contextual background, it is premature to state

EOD failed to change aging rate. Along the same line the author must also articulate what magnitude of changes they would have liked to see to conclude EOD is effective in reducing aging rate.

The method section does not accurately report when these extensive phenotyping was done. Obviously they could not be done in a single day or within a week. This is important for reproducibility as many phenotypes are sensitive to prior handling or stress. Similarly, many clinical and blood parameters show remarkable change between fasted and fed state even in mice under ad lib feeding condition. For these tests, the approximate time of the day when samples were collected should be included. Overall, an illustration showing the timeline of when those tests were done is essential for reproducibility and alternate interpretation. Similarly, for some of the tests, a standard practice is to habituate the animals to the instrument or protocol prior to collecting data. For those tests (examples include but not limited to indirect calorimetry, movement test, optokinetic response, etc) it is important to mention if the animals were habituated.

Some of the subpanels from supplementary materials should be moved to the main figure. Some of them include but are not limited to S-Fig. 1b, 1c, 4c, 5k, 5o, 5s, 6f, 6j, 6k. Additionally, the authors should include food consumption pattern over 47 h from indirect calorimetry measurements. At least S-Figure 4c, food consumption pattern must be included in the main figure.

Reviewer #2 (Remarks to the Author)

Review: Intermittent fasting extends murine lifespan but has limited effects on aging

Summary:

Xie and colleagues present a comprehensive analysis of the effects of IF (or more precisely alternate-day-feeding) on the health and lifespan of male C57BL/6 mice. Onset of IF at 2 months of age extended mean and maximum lifespan in male mice, which was likely mediated by a reduced number of neoplastic lesions per animal in the IF cohort. Notably, cancer remained the main cause of death for the control and IF cohort. The subsequent analysis of 239 phenotypes indicated that 116 showed aging-related traits of which only 7 were rescued in the aged IF cohort, and not the young IF cohort. However, IF positively affected 33 aging phenotypes in both the young and old IF cohort. The great majority of aging phenotypes were not altered by IF (67 out of 116), whereas 14 parameters were actually worsened by IF.

While the mechanistic insight into the life-extending effects of IF in male C57BL/6 mice remains unclear, the authors present a remarkably comprehensive study that investigates the effects of IF on a plethora of aging-related phenotypes.

Major

The authors write: "our data indicate that IF-induced lifespan extension in male C57BL/6J mice is sufficiently explained by a delay in lethal neoplastic disorders' There appear to be differences in the % cancer free survival and also in the % of mice with multiple tumors between IF and control diet mice. Are these differences not significant? If they are then, IF mice may have: 1) reduced tumor incidence, 2) reduced incidence of metastatic cancers. Since this is a central finding of the paper it will need an expanded and more careful interpretation.

I have only minor comments:

- Was the lifespan analysis and phenotyping carried out in two different cohorts? If so, please provide the numbers for each cohort. If the authors elected to start with one cohort, how were the animals selected for the each analysis (lifespan vs. phenotyping)?
- Please provide data for the food intake on feeding days. I assume the animals generally consumed more food on the refeeding days to compensate for the reduced calorie intake on the fasting days.
- All phenotyping tests were carried out at 24 months? What was the rationale for electing this

time point since some of the aging-related phenotypes occur later in life.

- Please provide p-values in the supplementary results. E.g. “we observed a possible slight improvement of latencies to fall ... without reaching statistical significance (p= XXX)”
- Why was the discussion on rapamycin included in the supplementary results? I don't think this contributes to clarify the study as only correlative comparisons between two studies can be made. If the authors propose rapamycin might provide a mechanistic understanding for the IF effects, the role of Tor should in the effects of IF should be investigated. In its current form, this is too vague.

Reviewer #3 (Remarks to the Author)

Review of “Intermittent fasting extends murine lifespan 1” but has limited effects on aging” by Xie et al. NCOMMS-16-25938-T

This is an important manuscript from the Dan Ehninger team. The authors present here a very important report on a carefully conducted characterization of phenotypes of aging in male mice. The authors put forth a tour the force study, a very interesting model and approach that will be helpful to others in the design of dietary manipulations aimed to modify health outcomes.

While I think the manuscript is novel and contains an abundance of data that would be useful to have published, in its current state it is hard to read/interpret, there are a few pieces of information/data that should be incorporated and discussed/clarified to further strengthen the document.

- 1.- What is the proper strain denomination? are these male C57BL6/J or C57BL6/NCrI? a rationale to the age, strain and sex for performing of these experiments should be clearly stated. It has to be noted and discussed the possibility that some of the phenotypes can have different trajectories from this intervention started in very early “adulthood” (2 month old) and, no data shown/collected at a middle time point (12-18 month old), thus we cannot be certain that some of the phenotypes did not differ in between those two time points nor that they will not change later in their lifespan.....as 24 month old in this colony of mice is right at 50% survival.
- 2.- The authors should discuss some of the findings in the context of the recent manuscripts on the effects of different degrees of CR in outcomes of health and lifespan in C57's, as well as in the context of the two ongoing non-human primate studies, in which there seem to be a differential response of a restriction intervention on outcomes of phenotypes of aging (health) and survival. Besides these, there are other manuscripts from Ingram and Goodrick (papers on behavioral characterization of this strain under, in their definition, intermittent feeding), and perhaps some of the same discussion points covered in Anson (AGE (2005) 27: 17–25), that should be also discussed

Point-by-point responses to reviewer comments

We have revised our manuscript according to the reviewers' comments as outlined below. Additionally, we reformatted the paper to fit Nature Communication's formatting requirements for articles as requested and, as a consequence, we expanded Introduction and Discussion and moved most of the supplementary material to the Results section and Figures of the main paper.

Reviewers' comments:

Reviewer #1 (Remarks to the Author):

The manuscript describes a dietary restriction regimen introduced on male C57B6 mice and its effects on different physiological outcomes. This is a large body of work done and most of the results are well presented. The experimental methods and the outcomes will also serve as a reference point for comparable studies. The authors should address the following points to strengthen the manuscript and to accurately represent the findings. Specifically, the authors must change the term IF to EOD for any publication in any journal.

We have changed the term IF to EOD throughout the revised manuscript as requested by the reviewer.

Similarly, the title is very misleading as it does not clearly articulate what would the authors expect if an intervention reduces aging (except making the mice immortal).

We would expect from an intervention slowing aging that it would exert preventative effects (or partial prevention) on a range of aging phenotypes in different tissues and physiological systems. We suggest the following revised title for the manuscript "Every-other-day feeding extends lifespan but fails to prevent many symptoms of aging in mice". This title accurately reflects our observation that lifespan extension occurred in EOD mice, but was associated with overall rather limited preventative effects on aging phenotypes in these mice (only 7 out of 116 aging phenotypes examined were prevented by EOD; see Fig. 1h in revised paper).

The experimental method clearly introduces two variables – prolonged fasting between periods of food access, and significant perturbation of normal sleep-wake cycle. One of them is beneficial, while the chronic perturbation of sleep wake cycle has adverse effect on health. In this context, it can be argued that the adverse effects of sleep-wake perturbation can be reduced by the fasting period that is integral to EOD. At the same time, it can also be argued that the maximum potential benefits of fasting may be reduced by the sleep-wake disturbances that are intrinsic to the protocol. This becomes an important confound in extending the protocol to a different diet composition that mimic the human diet preferences of moderately high fat or high sucrose content. This intrinsic confound in the study is a HUGE factor in the accurate interpretation and future extension of study findings. While it is impractical to repeat any of the experiments, the authors must put some time into re-writing several parts of the manuscript to highlight and discuss this point.

We have added a discussion of this topic to the revised paper (see page 25 of the revised manuscript):

"Under conditions of food shortage, nocturnal rodents, such as mice, are known to shift their activity pattern to also cover parts of the light phase^{63,64}; this could represent an adaptive response that increases the chance to acquire food that is available only during restricted temporal windows⁶³. Accordingly, altered circadian activity patterns have to be taken into account as potential confounds when considering outcomes of dietary restriction (DR) studies, including those employing EOD and CR^{64,65}. However, prior analyses, using CR, showed that feeding time influenced circadian rhythms, but did not affect CR-induced lifespan extension^{66,67,68}. Moreover, ad libitum-fed mice subjected to weekly 12-hour shifts of their light-dark-schedule during much of their lifetime did not show alterations

in life expectancy compared to controls⁶⁷. These data support the notion that DR effects on circadian rhythms are independent of longevity effects. Additional studies are needed to clarify whether DR-induced circadian alterations might interact with DR effects on aging phenotypes other than age-related mortality.

Our metabolic profiling experiments indicated that longevity in EOD mice was not associated with reduced, but rather slightly increased rates of energy expenditure, as evidenced by body weight-adjusted oxygen consumption, carbon dioxide emission and heat production. Based on the circadian pattern of food intake and RER values (though not necessarily overall locomotor activity), our metabolic analyses also provided some evidence for the expected alterations in circadian activity patterns in EOD mice (see above), which were particularly evident in the young cohort of animals. These changes were less clear in aged mice on almost lifelong EOD, raising the possibility that DR-associated shifts in circadian activity and associated metabolic parameters may be transient in nature and return to more typical nocturnal patterns with extended exposure to DR.”

Suggested new title "Every other day food intake extends murine lifespan, but has variable effects on markers of aging"

We appreciate the suggestion, which we would propose to modify slightly to better account for the main finding that EOD extended life but prevented few aging traits: “Every-other-day feeding extends lifespan but fails to prevent many symptoms of aging in mice”.

Major concern. The title of the manuscript should accurately reflect the procedure used. With different dietary restrictions, the term intermittent fasting is loosely used to cover many types of fasting regimens. The range of fasting regimens differs in the duration of fasting, caloric intake and have diverse significance for reduction to practice in humans. The authors must clearly state they tried Every Other Day feeding (EOD) in the abstract and change to title to reflect it is an EOD protocol. All figure panels, tables and in text reference to “intermittent fasting” must be replaced with EOD to reflect the factually correct procedure used in the manuscript.

We have changed the terminology throughout the manuscript as suggested by the reviewer.

The fasting method refers to a protocol used more than a quarter century old when researchers did not pay any attention to the natural eating and sleeping pattern of mice. Providing food to nocturnal mice at 9am local time forces them to stay awake and consume the food, which can perturb their natural cycle of feeding-fasting and sleep-wakefulness.

This disruption is clearly evident in supplementary figure 4c. Essentially the EOD forces the mice to be awake and feed for almost 24 h on the day of feeding. Therefore, EOD is a combination of both 24 h fast every other day and a shiftwork like regimen. I strongly suggest the author place Figure 4c in the main figure to show the daily metabolic pattern in the cohorts. This 24 h RER is one of the important physiological phenotypes in which these cohorts differed and it should be clearly shown in the main text. With numerous papers in both animal and human models showing this diurnal perturbation can have adverse metabolic and neoplastic consequences, the conclusion of the manuscript does include a large confound. Therefore, it is inaccurate to conclude that the procedure had limited impact on aging.

As mentioned above, we have added the following discussion to the revised paper (see page 25 of the revised manuscript and below). The indirect calorimetry results were moved to the main part of the paper.

“Under conditions of food shortage, nocturnal rodents, such as mice, are known to shift their activity pattern to also cover parts of the light phase^{63,64}; this could represent an adaptive response that increases the chance to acquire food that is available only during restricted temporal windows⁶³. Accordingly, altered circadian activity patterns have to be taken into account as potential confounds when considering outcomes of dietary restriction (DR) studies, including those employing EOD and CR^{64,65}. However, prior analyses, using CR, showed that feeding time influenced circadian rhythms, but did not affect CR-induced lifespan extension^{66,67,68}. Moreover, ad libitum-fed mice subjected to

weekly 12-hour shifts of their light-dark-schedule during much of their lifetime did not show alterations in life expectancy compared to controls⁶⁷. These data support the notion that DR effects on circadian rhythms are independent of longevity effects. Additional studies are needed to clarify whether DR-induced circadian alterations might interact with DR effects on aging phenotypes other than age-related mortality.

Our metabolic profiling experiments indicated that longevity in EOD mice was not associated with reduced, but rather slightly increased rates of energy expenditure, as evidenced by body weight-adjusted oxygen consumption, carbon dioxide emission and heat production. Based on the circadian patterns of food intake and RER values (though not necessarily overall locomotor activity), our metabolic analyses also provided some evidence for the expected alterations in circadian activity patterns in EOD mice (see above), which were particularly evident in the young cohort of animals. These changes were less clear in aged mice on almost lifelong EOD, raising the possibility that DR-associated shifts in circadian activity and associated metabolic parameters may be transient in nature and return to more typical nocturnal patterns with extended exposure to DR."

The manuscript is presented with no clear focus on why some of the phenotypes were measured. It sounds like a list of capabilities the PIs have access to.

We have added the following clarification to the revised paper (see page 6):

"The parameters selected corresponded in large part to the ones used in our prior large-scale analysis of rapamycin's effects on aging in male C57BL/6J mice¹³."

For each phenotype measured, the author should include some information on how that specific phenotype changes with age, whether there is any evidence that any prior intervention has improved that specific age-dependent change in the phenotype and then make the interpretation whether this EOD intervention was effective or not in reducing the age-dependent change. This can be presented as an additional column in some of the supplementary tables. Without such clear contextual background, it is premature to state EOD failed to change aging rate.

We have added, throughout the main text, references to prior studies on how specific phenotypes are expected to change during aging in mice (see Results section of the revised paper). The following statement was also added to the revised manuscript (see page 6):

"The parameters selected corresponded in large part to the ones used in our prior large-scale analysis of rapamycin's effects on aging in male C57BL/6J mice¹³. We provide additional references, throughout the main text, indicating that many of the phenotypes analyzed represent robust aging traits that are reliably observed across a range of studies in mice."

Along the same line the author must also articulate what magnitude of changes they would have liked to see to conclude EOD is effective in reducing aging rate.

As mentioned above, we would expect from an intervention slowing aging that it would exert preventative effects (or partial prevention) on a range of aging phenotypes in different tissues and physiological systems. If the majority of detected age-related traits were prevented partially or fully by EOD, we would conclude that EOD is an effective intervention in reducing aging rate. However, only 7 out of 116 aging phenotypes examined displayed preventative effects on aging phenotypes (see Fig. 1h in revised paper).

The method section does not accurately report when these extensive phenotyping was done. Obviously they could not be done in a single day or within a week. This is important for reproducibility as many phenotypes are sensitive to prior handling or stress. Similarly, many clinical and blood parameters show remarkable change between fasted and fed state even in mice under ad lib feeding condition. For these tests, the approximate time of the day when samples were collected should be included. Overall, an illustration showing the timeline of when those tests were done is essential for reproducibility and alternate interpretation. Similarly, for some of the tests, a standard practice is to habituate the animals to the instrument or protocol prior to collecting data. For those tests (examples

include but not limited to indirect calorimetry, movement test, optokinetic response, etc) it is important to mention if the animals were habituated.

We have added the additional information requested by the reviewer (see below).

- We specified the individual time periods during which the different phenotyping components were performed:

“The following analyses were performed (in the order mentioned)^{73,74}: open field (week 1), modified SHIRPA (week 1), grip strength (week 1), rotarod (week 2), acoustic startle response and pre-pulse inhibition (PPI) (week 2), hot plate test (week 4), transepidermal water loss test (week 4), indirect calorimetry (week 5), NMR-based body composition analysis (week 6), glucose tolerance test (week 7), awake electrocardiography and echocardiography (week 8), Scheimpflug imaging (week 9), optical coherence tomography (week 9), laser interference biometry (week 9), virtual drum vision test (week 9), clinical chemistry (week 10), hematology (week 10), FACS analysis of peripheral blood leukocytes (week 10), Bioplex ELISA (Ig concentrations) (week 10), auditory brain stem response (week 11), X-ray/bone densitometry (week 11) and pathology (week 13).”

- We provide additional information regarding blood sample collection:

For fed-state clinical chemistry analyses, we collected blood sample of EOD and AL mice in the morning of a EOD-group fasting day immediately after food withdrawal (i.e., EOD mice have had 24 hours ad libitum access to food prior to blood collection; ad libitum controls have, as usual, also had free access to food).

For fasted-state clinical chemistry analyses, blood samples of EOD and AL mice were collected on an EOD-group fasting day 6 hours after the food withdrawal (i.e., both EOD and AL mice were fasted for 6 hours prior to blood collection; after blood collection each group was continued with its assigned feeding regime - that is ad libitum access to food in case of AL mice and continued fasting in EOD mice).

A few clinical chemistry parameters were analyzed in both the fasted and fed state (e.g. plasma cholesterol and triglycerides). While the results of the fasted-state and fed-state clinical chemistry analyses cannot be compared directly (because these analyses were not run side-by-side), it is of course expected that some measures differ between fed and fasted state in the AL group.

- We provided additional information regarding habituation to test environments:

General remark (page 28 of the revised manuscript): “Mice (in their home cages) were habituated to the test room for a period of 15 min prior to commencement of analyses for most experimental procedures (unless stated otherwise).”

Open field (page 30 of the revised manuscript): “Animals were transported to an area directly adjacent to the testing room 30 min prior to open field analyses. The 20-min open field test itself was then started right away given that the novelty aspect of the environment is a crucial component of the test.”

Acoustic startle reflex (ASR) and pre-puls inhibition of ASR (page 30 of the revised manuscript): “Animals were transported to an area directly adjacent to the testing room 30 min prior to behavioral analyses. Next, we employed a 5 min acclimation period to the test compartment (i.e., a mouse restrainer), before the actual testing began.”

Indirect calorimetry (page 32 of the revised manuscript): “Mice were placed individually into these respirometry cages for about 2 hours before the actual recordings were started. We carefully monitored that mice successfully used food hoppers and water bottles during this habituation period and also throughout the consecutive 47 hours.”

Some of the subpanels from supplementary materials should be moved to the main figure. Some of them include but are not limited to S-Fig. 1b, 1c, 4c, 5k, 5o, 5s, 6f, 6j, 6k. Additionally, the authors

should include food consumption pattern over 47 h from indirect calorimetry measurements. At least S-Figure 4c, food consumption pattern must be included in the main figure.

We reformatted the paper to fit Nature Communication's formatting requirements for articles and moved most of the material to main figures in the revised manuscript.

Reviewer #2 (Remarks to the Author):

Major

The authors write: "our data indicate that IF-induced lifespan extension in male C57BL/6J mice is sufficiently explained by a delay in lethal neoplastic disorders' There appear to be differences in the % cancer free survival and also in the % of mice with multiple tumors between IF and control diet mice. Are these differences not significant? If they are then, IF mice may have: 1) reduced tumor incidence, 2) reduced incidence of metastatic cancers. Since this is a central finding of the paper it will need an expanded and more careful interpretation.

We calculated the results of the corresponding statistical comparisons as requested. The differences in % of cancer free survival (Fisher's exact test, $p=0.2445$) as well as in % of multiple tumor incidence (Fisher's exact test, $p=0.0893$) were statistically not significant between EOD and AL fed animals. Overall tumor burden at death, however, was significantly reduced in EOD mice compared to controls (average number of tumors per mouse: t-test, $p=0.0308$)

Was the lifespan analysis and phenotyping carried out in two different cohorts? If so, please provide the numbers for each cohort. If the authors elected to start with one cohort, how where the animals selected for the each analysis (lifespan vs. phenotyping)?

All aging animals used for lifespan analysis and phenotyping assessments were raised as one cohort. Prior to the commencement of phenotyping, animals that were housed together in individual cages were then in part assigned to the phenotyping, in part to the survival analysis. Additionally, we made sure that overall body weight distributions were matched between animals that were assigned to phenotyping or survival analysis. Two months old young mice were assigned to either AL or EOD dietary regimen one month prior of initiation of phenotyping analyses. Young and aged animals assigned to phenotyping were analyzed side-by-side in all experiments.

Please provide data for the food intake on feeding days. I assume the animals generally consumed more food on the refeeding days to compensate for the reduced calorie intake on the fasting days.

Food intake data are presented in Fig. 1b of the revised paper. We measured food intake once per month over the entire course of the study. During these once monthly measurements we obtained data on food consumption over four consecutive days covering two fasting and two feeding days (see page 28). As expected⁶², animals learned to gorge whenever food was available, resulting in an overall reduction of average calorie intake over the entire lifespan by only about 7.5% compared to ad libitum fed controls. We conclude that the calorie deficit caused by fasting periods was almost fully compensated for by increased food intake on feeding days.

All phenotyping tests were carried out at 24 months? What was the rationale for electing this time point since some of the aging-related phenotypes occur later in life.

Our strategy was to evaluate phenotypes at an age where many aging traits are evident, yet only relatively few of the mice have died.

We added a paragraph to the revised paper that discusses these limitations (see pages 25 and 26):

"Our study had several limitations: The data were collected in one specific mouse strain (i.e., C57BL/6J) and sex (i.e., male). We focused our analyses on two specific age groups, including a young cohort with dietary restriction beginning at 8 weeks of age and phenotypic analyses commencing at approx. 3 months of age, as well as an aged cohort with dietary restriction commencing at 8 weeks of age and phenotyping starting at ca. 21 months of age. We did not examine additional intermediate or older cohorts. Future studies including such additional groups could define more detailed lifetime trajectories of the different traits examined. Furthermore, we would like to note that additional studies are needed to address whether other methods of dietary restriction, such as CR, have broader effects on the prevention of natural aging."

Please provide p-values in the supplementary results. E.g. “we observed a possible slight improvement of latencies to fall ... without reaching statistical significance ($p = XXX$)”

We added p-values throughout the Results section where appropriate.

Why was the discussion on rapamycin included in the supplementary results? I don't think this contributes to clarify the study as only correlative comparisons between two studies can be made. If the authors propose rapamycin might provide a mechanistic understanding for the IF effects, the role of Tor should in the effects of IF should be investigated. In its current form, this is too vague.

We have removed this section in the revised version of the paper as suggested by the reviewer.

Reviewer #3 (Remarks to the Author):

What is the proper strain denomination? are these male C57BL6/J or C57BL6/NCrl?

The proper strain denomination of the mice used is C57BL/6J. These mice were purchased from Charles River Laboratories (CRL) because CRL is the exclusive distributor of Jackson Lab mice (including C57BL/6J) in Germany.

A rationale to the age, strain and sex for performing of these experiments should be clearly stated.

Our choice regarding age, strain, sex and other variables was guided by the attempt to mimic, as closely as possible, the conditions used in a prior study that described longevity effects of every-other-day feeding in mice.

This is specified in more detail in the Methods section of the revised paper as follows (see page 29):

“We chose a dietary restriction regimen (i.e., EOD) previously shown to extend lifespan in mice¹¹ and compatible with group housing the animals. Unlike other dietary restriction paradigms (such as restriction of calorie intake to 60% of what animals would eat ad libitum), the feeding situation in the EOD paradigm is not associated with competition for food (because food is available in excess during the feeding day) and therefore does not require single housing of the animals.

Animals fed ad libitum were granted unlimited access to food anytime. EOD was conducted according to a published protocol allowing free access to food for 24 hours alternated by food deprivation for 24 hours (also referred to as every-other-day feeding)¹¹. Food was provided to the EOD cohorts at 9 am and withdrawn at 9 am on the next morning as described previously¹¹. EOD was initiated at 2 months of age, corresponding to the age of EOD onset associated with the largest effect size on lifespan in a previous study on male C57BL/6J mice¹¹. All animals received a standard rodent chow (Altromin 1314; closely matching the composition of the rodent chow used in a previous EOD longevity study on male C57BL/6J mice¹¹ and were kept on their respective diet regimen for the entirety of the study. The Altromin 1314 chow came in solid pellets. Careful pilot analyses showed that mice did not crumble these pellets. Accordingly, removing the pellets on the restriction days was sufficient to fully deprive the animals of food (no cage change required).”

It has to be noted and discussed the possibility that some of the phenotypes can have different trajectories from this intervention started in very early “adulthood” (2 month old) and, no data shown/collected at a middle time point (12-18 month old), thus we cannot be certain that some of the phenotypes did not differ in between those two time points nor that they will not change later in their lifespan.....as 24 month old in this colony of mice is right at 50% survival.

We appreciate this comment and added a paragraph to the revised paper that discusses these limitations (see pages 25 and 26):

“Our study had several limitations: The data were collected in one specific mouse strain (i.e., C57BL/6J) and sex (i.e., male). We focused our analyses on two specific age groups, including a young cohort with dietary restriction beginning at 8 weeks of age and phenotypic analyses commencing at approx. 3 months of age, as well as an aged cohort with dietary restriction commencing at 8 weeks of age and phenotyping starting at ca. 21 months of age. We did not examine additional intermediate or older cohorts. Future studies including such additional groups could define more detailed lifetime trajectories of the different traits examined. Furthermore, we would like to note that additional studies are needed to address whether other methods of dietary restriction, such as CR, have broader effects on the prevention of natural aging.”

The authors should discuss some of the findings in the context of the recent manuscripts on the effects of different degrees of CR in outcomes of health and lifespan in C57's, as well as in the context of the two ongoing non-human primate studies, in which there seem to be a differential response of a restriction intervention on outcomes of phenotypes of aging (health) and survival.

We have added a discussion of this topic to the revised paper (see pages 23, 24 and 26):

“Dietary restriction regimens commonly used in rodents are EOD (employed in the present study) and CR. One of the important differences between these two protocols is the net calorie intake. Daily food supply of animals assigned to chronic CR is usually cut down to 60% of the amount consumed ad libitum by age-matched controls (equivalent with 40% restriction). Chronic CR results in lifespan extension and pronounced growth retardation such that body weight (-42% in rat, -35% in mouse), fat mass (-70% in rat), and many organs weights (heart: -29% in rat; liver: -34% in rat, -31% in mouse; kidney: -33% in rat, -10% in mouse; spleen: -50% in rat, -66% in mouse; prostate: -25% in rat) are overall decreased in adulthood^{6,8,9}. Exceptions are sizes of brain and testis, which remain unaffected by chronic CR⁹.

In contrast, animals subjected to EOD quickly adjust feeding to times of food availability and, therefore, show more modest reductions in net calorie intake compared to CR. In line with the CR findings mentioned above, EOD had strongest effects on organ weight with regards to the spleen (-31% in young, -56% in old). However, body length, fat mass and the weights of most organs were affected more modestly by EOD. Thus, when compared to CR, EOD avoids partially the CR-associated growth retardation, while still affording the benefit of lifespan extension and other health benefits, such as improved insulin sensitivity and hippocampal neuroprotection against excitotoxic injury^{4,11,52}.”

*“Two important studies using rhesus macaques (*Macaca mulatta*) were launched in the late 1980s to explore CR-driven long-term survival and health outcomes in primates. The first study was initiated at the National Institute on Aging (NIA) in 1987, the latter started at the Wisconsin National Primate Research Center (WNPRC) in 1989. While CR extended lifespan in the WNPRC cohort, survival was not improved in the NIA study^{69,70}. This discrepancy in survival outcomes might be attributable to differences in study design, including feeding protocols (e.g., control monkeys in the WNPRC study ate ad libitum, while controls in the NIA study received defined daily food portions)⁷¹. The incidence of specific age-related diseases (including cancer and diabetes) was reduced by CR in both the WNPRC and NIA study, showing that some of the expected health benefits are reproducible across studies. The impact of EOD in non-human primates has not been investigated so far and remains to be determined in future studies.”*

Besides these, there are other manuscripts from Ingram and Goodrick (papers on behavioral characterization of this strain under, in their definition, intermittent feeding), and perhaps some of the same discussion points covered in Anson (AGE (2005) 27: 17–25), that should be also discussed.

We have added a discussion of these studies, as well as additional work on every-other-day feeding to the revised version of the manuscript (see page 24):

“Prior EOD studies had examined restriction effects on a few additional parameters besides lifespan. Consistent with our observation of elevated spontaneous locomotor activity in EOD mice (in an open field assay), previous experiments in rats had found increased levels of voluntary wheel running under EOD restriction⁵⁹. EOD exerted cardioprotective effects in rodent ischemia models and, in line with the observations in the present study, influenced a number of cardiac dimensional measures consistent with a retardation of heart growth in restricted animals⁶⁰. We also confirmed prior results regarding expected reductions of plasma glucose, triglyceride and insulin concentrations in rodents subjected to EOD^{52,61,62}.”

Reviewers' Comments:

Reviewer #2:

Remarks to the Author:

The authors have addressed all of my concerns. I wonder whether in the title it may be more correct to say "fail to delay" rather than "fail to prevent". This would have to be based on whether it is reasonable to expect that the majority of symptoms listed would be prevented completely versus simply delayed.

Point-by-point responses to reviewer comments

We thank you very much for the work you put in the present manuscript. In the revised manuscript (second revision), we have implemented the editorial changes requested and have addressed reviewer #2's comment as outlined below.

Reviewer #2 (Remarks to the Author):

The authors have addressed all of my concerns. I wonder whether in the title it may be more correct to say "fail to delay" rather than "fail to prevent". This would have to be based on whether it is reasonable to expect that the majority of symptoms listed would be prevented completely versus simply delayed.

We have altered the title as suggested by the reviewer. The new title of the manuscript is: Every-other-day feeding extends lifespan but fails to delay many symptoms of aging in mice.